# A large language model for clinical outcome adjudication from telephone follow-up interviews: a secondary analysis of a multicenter randomized clinical trial

Zhao Shi [1,6], Bingqian Wu[2,6], Bin Hu[1,6], Jian Zhong[1,6], Zezhong Li[3], Fandong Zhang[3], Zijian Chen[1], Chun Yang[1], Bangjun Guo [1], Qinmei Xu[4], Huimin Pang[1], Han Wang[1], Yueyan Wang[1], Jinlong Zhao[1], Jing Xu[1], Yizhou Yu [5] & Long Jiang Zhang [1,2] ✉

Automated adjudication of clinical outcomes from telephone follow-ups is crucial for reducing workload and increasing data quality in large-scale trials. Here, we show that a domain-specific large language model (Fu-LLM) effectively automates the preadjudication of key clinical events—including death, hospitalization, and medication use—based on 1,046 vignettes of follow-up telephone interviews conducted across three centers in a randomized clinical trial (China CT-FFR Study 3). Fu-LLM outperforms not only state-of-the-art general-purpose LLMs (e.g. GPT-3.5-turbo, GPT-4o, DeepSeek-v3, Claude 3.5-Sonnet, and Gemini-2.0-Pro) and conventional machine learning models (Support Vector Machine), but also human adjudicators in a silico human −model comparative study. It also shows greater robustness than different versions of GPT-4 do in temporal drift tests. Our findings demonstrate that Fu-LLM can significantly streamline outcome identification in clinical trials, offering a scalable and accurate tool for automating labour-intensive adjudication processes.

Clinical trials are key to evaluating the development of safe, dependable, and efficient interventions, and longitudinal follow-up is paramount for determining endpoint[1]. Telephone-based follow-up remains the predominant modality in multicentre trials, because of its capacity for the rapid uptake of information and communication[2]. which has been demonstrated to yield comparable or lower levels of missing data, increased follow-up rates, and enhanced clinical outcomes[3–6]. However, conventional manual adjudication incurs substantial operational burdens, including transcription and endpoint verification, which are not only time-consuming and labor-intensive but also pose

risks of human error and the omission of key information[7,8]. Automated adjudication of critical clinical outcomes from telephone follow-up interviews is needed to reduce workload and improve follow-up quality, and to potentially reduce costs, especially for large-scale clinical trials.

Despite these advances, machine learning (ML) approaches for this task remain challenging, prompting the development of novel technical solutions. Interactive voice response systems (IVRSs) have been utilized for telephone-based follow-up, including assessment after emergency department adverse events and screening for

[1]Department of Radiology, Jinling Hospital, Affiliated Hospital of Medical School, Nanjing University, Nanjing, China. [2]Department of Radiology, Jinling Hospital, Nanjing Medical University, Nanjing, China. [3]AI lab, Deepwise Healthcare, Beijing, China. [4]Stanford Center for Biomedical Informatics Research (BMIR), Department of Medicine, Stanford University, Stanford, CA, USA. [5]Department of Computer Science, The University of Hong Kong, Hong Kong, China. [6]These authors contributed equally: Zhao Shi, Bingqian Wu, Bin Hu, Jian Zhong. ✉e-mail: kevinzhlj@163.com

potentially drug-related symptoms[9,10]. However, IVRS lack contextual understanding of clinical narratives. Transformative natural language processing (NLP) models based on transformer architectures have been developed to adjudicate heart failure hospitalizations[11,12]; however, these models are restricted to structured clinical documentation (e.g., discharge summaries) and cannot process unstructured conversational data.

Large language models (LLMs) have demonstrated potential for clinical text interpretation[13]. and have been trained extensively on textual data in an unsupervised manner to infer linguistic relationships[13,14]. They have shown great potential for transforming health-care delivery, and clinicians have begun to use LLMs to communicate with patients and draft clinical notes[15–17]. However, compared with structured medical records, LLMs exhibit reduced accuracy when conversational data are used[18] and face critical challenges, such as clinical domain errors, which include hallucinations[19] and omissions from medical document summarization[20,21]; temporal instability, which indicates that short- and long-term temporal variations in LLM output exist[22]; and validation gaps, which indicatesa lack of sufficient benchmarking against ground references. Given the rapidly evolving landscape of LLMs and the potential for both transformative advancements and unintended consequences, rigorous studies are warranted to develop domain-specific fine-tuning LLMs with clinical dialog augmentation and rigorous examine their clinical effectiveness in terms of clinical outcome adjudication in follow-up telephone interviews.

Therefore, the purpose of this secondary analysis study is to develop a domain-specific follow-up large language model (Fu-LLM), investigate its feasibility for automatically adjudicating participants' clinical outcomes through telephone dialog, including information sources, death, hospitalization, invasive coronary angiography (ICA) and medication events in a multicentre, prospective randomized clinical trial (China CT-FFR study 3) and further evaluate the effectiveness of Fu-LLM by conducting a series of benchmark comprehensive baseline comparisons against other popular public LLMs, traditional ML models, and human adjudicators.

## Results

### Study design and participants
This study is a secondary analysis of the China CT-FFR study 3, which is a multicentre, prospective randomized clinical trial (ChiCTR.org.cn Identifier: ChiCTR2100053219), and the 1-year outcome was published[23]. The ethics committee at each participating center approved the trial protocol. To guide downstream care at 17 tertiary hospitals across China from May 1 to September 30, 2021, 5410 participants with suspected coronary artery disease were randomly assigned to either a group in which initial testing was performed with fully-automated coronary CT angiography (CCTA)-derived fractional flow reserve (CT-FFR) and CCTA (CCTA + CT-FFR, $n = 2704$) or to group in which CCTA alone was performed ($n = 2706$). The primary endpoint of the study was the rate of ICA within 90 days after CCTA was performed, which was adjudicated by an independent clinical events committee (CEC) using medical records (admission notes or discharge summaries at a minimum). Three centers (Jinling Hospital [JL Center], Nanjing First Hospital [NFH Center], and the affiliated hospital of Jining Medical University [JNU Center]) whose sound recordings of 1-year follow-up telephone interviews were saved were included in this secondary study. The flowchart of participant enrollment is shown in Fig. 1.

A total of 1191 vignettes of telephone interviews of follow-up were collected from the three hospitals, of which 1046 were included for the final analysis (Fig. 1), representing 1022 participants (mean [standard deviation] age, 63.1 [11.0] years; 403 [39.4%] females; 619 [60.6%] males). Participants were assigned to either the CCTA alone group ($n = 498$, 48.7%) or CCTA + CT-FFR group ($n = 524$, 51.3%). The median interquartile range (IQR) duration of the recording was 147 (96–216) seconds. Since the LLMs accepted text-only input and lacked publicly available multimodal capabilities during the study period, the recordings were converted to text using iFlytek's voice recognition technology. These text-based conversations served as model inputs. All the conversions occurred on HIPAA-compliant servers. Deidentified text was used for model training following institutional ethics approval. The median conversation length was 587.5 Chinese characters (IQR 362.8–865.5). A total of 321 recordings from 307 participants (30.0%) were obtained for the JL Center, 355 recordings from 345 participants (33.8%) were obtained for the NFH Center, and 370 recordings from 370 participants (36.2%) were obtained for the JNU Center. The primary endpoint of the ICA was reached for 112 participants (11.0%) as determined by the CEC (CCTA group: 58 [11.6%]; CCTA + CT-FFR group: 54 [10.3%]). Baseline characteristics and endpoint data are detailed in Table 1.

### Silver reference standards
The first step in outcome judgment is telephone interviews before adjudication, on the basis of which hospitalization and outpatient visit information during participants' follow-up can be further traced and collected for verification of participants' self-reported events. Therefore, experts from the core laboratory and the CEC of China CT-FFR study 3 determined telephone interview outcome classifications into five categories: (1) whether the information came from the participant themselves; (2) whether the participant died during the follow-up period; (3) whether the participant was hospitalized during the follow-up period; (4) whether the participant underwent ICA during the follow-up period; and (5) whether the participant took medication during the follow-up period. For deceased participants, questions regarding hospitalization, surgery, and medication were omitted. Since participants might not understand "ICA," follow-up staff used "surgery" during interviews, with ICA verified through medical records. The panel established adjudication criteria based exclusively on call recordings using four labels: "yes", "no", "uncertain", or "not mentioned" (Supplementary Method 1). Three follow-up professionals (J.Z., Z.C. and C.Y.) set the silver reference standards ("Methods").

In accordance with to the silver reference standards, participant self-reporting occurred in 59.4% of participants (622/1046), and 2.1% (22/1046) of the participants died; regarding hospitalization, 20.3% (208/1024) answered "yes", 16.1% (165/1024) were uncertain, and 8.3% (85/1024) met the "not mentioned" criteria. Regarding whether surgery was performed, 16.7% (171/1024) answered "yes", 0.5% (5/1024) were uncertain, and 31.0% (317/1024) met the "not mentioned" criteria. Regarding whether medication was taken during the follow-up period, 80.8% (827/1024) answered "yes", 1.6% (16/1024) were uncertain, and 3.5% (36/1024) met the "not mentioned" criteria.

### Development of a domain-optimized LLM-based model (Fu-LLM) for outcome preadjudication
We conducted supervised fine-tuning (SFT) based on the Qwen2-7B model to develop the Fu-LLM, which can automatically extract five specified clinical outcomes from follow-up dialogs. The 1046 real-world samples were evaluated using fivefold cross-validation. Given the limited dataset size, two data augmentation strategies (data rewrite and data synthesis) were adopted to increase the data volume and diversity, and post-processing techniques (data deduplication and content filtering) were employed to further improve the quality of the generated data. These strategies yielded 19,162 valid data entries, with 2439 originating from the data re-write strategy and 16,723 entries originating from the data synthesis strategy. We employed SFT combined with low-rank adaptation (LoRA)[24] to fine-tune the Qwen2-7B model. The model was trained with a maximum sequence length of 4096 tokens and 8-step gradient accumulation. Training was conducted on 6 NVIDIA H20 GPUs with a total batch size of 48. The

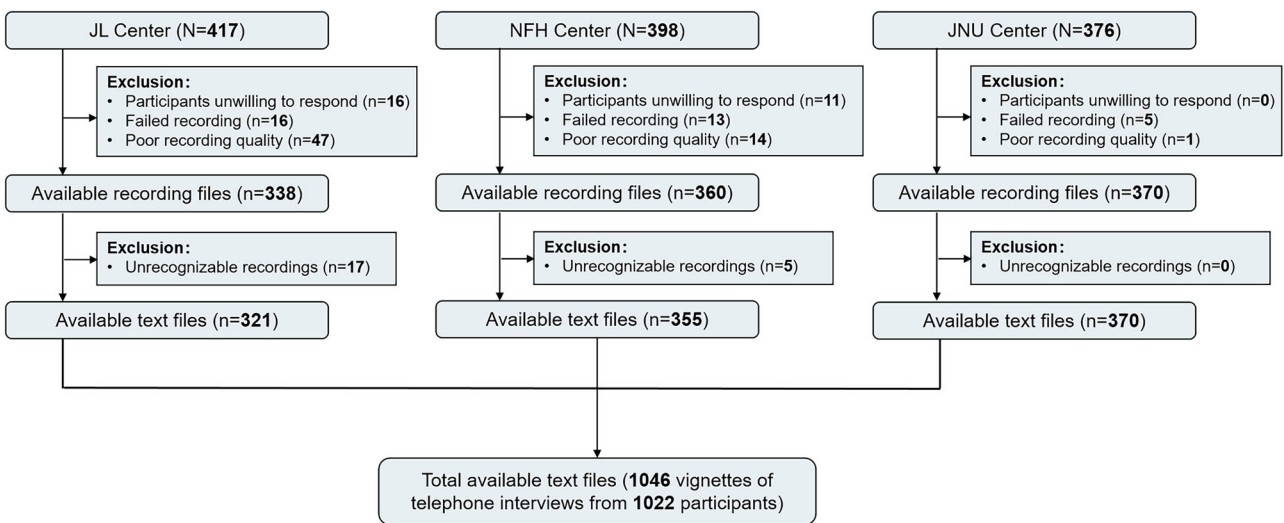

**Fig. 1 | Overview of the data composition of participants enrollment.** JL Jinling Hospital; JNU the affiliated hospital of Jining Medical University; NFH Nanjing First Hospital. *N* and *n* are the number of vignettes of telephone interviews. Source data are provided as a Source data file.

AdamW optimizer was used with an initial learning rate of $1 \times 10^{-4}$, a cosine learning rate scheduler, and a warm-up ratio of 5% to stabilize the early training phase. The model was trained for 3 epochs to balance stability and performance ("Methods"). Finally, three versions of the Fu-LLM were developed for evaluation (examples are presented in Supplementary Method 3):

- finetune_qwen2_7b, fine-tuned using all the training datasets, including the augmented samples;
- finetune_qwen2_7b_wo_aug, fine-tuned using the original data without data augmentation;
- zero_shot_qwen2_7b, directly utilized the pretrained Qwen2-7B model without any task-specific fine-tuning.

The researchers input the dialog texts individually for each center. Instruction prompt engineering was applied to adapt the chatbot verbatim for clinical outcome adjudication and to output "yes", "no", "uncertain", or "not mentioned" classifications for the five outcomes. The finetune_qwen2_7b model performed best, achieving the highest rate of raw agreement (93.7% [95% confidence intervals (CI), 93.1–94.3], a Cochran's $Q$ value of 1378.8, $p < 0.001$), which was significantly greater than that of finetune_qwen2_7b_wo_aug (88.9% [95% CI, 88.1–89.7], adjusted $p < 0.001$) and zero_shot_qwen2_7b (72.5% [95% CI, 71.3–73.7], adjusted $p < 0.001$). For silver reference adjudication ("yes"/"no" classifications), finetune_qwen2_7b demonstrated higher sensitivity than zero_shot_qwen2_7b (97.5% [95% CI, 96.7–98.2] vs. 82.1% [95% CI, 80.2–83.6], adjusted $p < 0.001$) and the finetune_qwen2_7b_wo_aug model (94.4% [95% CI, 93.3–95.4], adjusted $p < 0.001$) did. Its specificity (95.0% [95% CI, 94.2–95.8]) exceeded those of zero_shot_qwen2_7b (70.3% [95% CI, 68.5–71.9], adjusted $p < 0.001$) and finetune_qwen2_7b_wo_aug (92.1% [95% CI, 91.1–93.1], adjusted $p < 0.001$). Notably, the negative predictive value (NPV) of finetune_qwen2_7b (98.2% [95% CI, 97.8–98.7]) surpassed those of zero_shot_qwen2_7b (85.1% [95% CI, 83.5–86.5], $p < 0.05$) and finetune_qwen2_7b_wo_aug (96.0% [95% CI, 95.2–96.7], $p < 0.001$) (Table 2). Confusion matrices for each outcome of Fu-LLM (finetune_qwen2_7b) and the silver references are shown in Fig. 2. The center-specific results are shown in Supplementary Table 1.

### Risk factors of Fu-LLM for outcome preadjudication

The risk factors affecting Fu-LLM (finetune_qwen2_7b) during outcome preadjudication were analysed for each event. With respect to information source adjudications, multivariable analysis revealed that participants who smoked at baseline (adjusted odds ratio [aOR] = 3.0;

$p = 0.008$) and those with baseline chest stuffiness symptoms (aOR = 0.3; $p = 0.015$) were associated with Fu-LLM concordance with silver references. With respect to death event adjudications, multivariable analysis revealed no significant associations with Fu-LLM concordance. With respect to hospitalization event adjudications, the JNU Center was an independent risk factor (aOR = 2.7, $p < 0.001$). With respect to surgery event adjudications, only participants who required urgent revascularization during follow-up were associated (aOR = 11.3; $p = 0.030$). For medication use adjudications, the independent risk factors included the treatment group (aOR = 2.8; $p = 0.004$), baseline smoking status (aOR = 0.3; $p = 0.032$), and baseline CAD-RADS 2 status (aOR = 4.5; $p = 0.016$). Neither conversation text length nor duration were independent risk factors for any preadjudication outcome (all $p > 0.05$).

We also performed a subgroup analysis to assess whether group assignment (CCTA vs. CCTA + CT-FFR) affected Fu-LLM performance. The results demonstrated that the raw agreement (93.4% [95% CI, 92.4–94.4] vs. 93.9% [95% CI, 93.0–94.8], $p = 0.441$), sensitivity (96.9% [95% CI, 95.7–98.0] vs. 98.1% [95% CI, 97.2–98.9], $p = 0.090$), specificity (94.9% [95% CI, 93.7–96.1] vs. 95.1% [95% CI, 94.0–96.2], $p = 0.839$), positive predictive value (PPV) (92.9 [95% CI, 91.2–94.5] vs. 93.3 [95% CI, 91.8–94.9], $p = 0.692$) and NPV (97.8 [95% CI, 97.0–98.6] vs. 98.6 [95% CI, 98.0–99.2], $p = 0.114$) did not significantly differ between the two groups (Supplementary Table 2).

There are several patterns for the samples for which Fu-LLM yields incorrect predictions (examples are shown in Supplementary Method 4). First, overly long dialogs hinder information processing, whereas very brief dialogs provide insufficient detail. Second, event timing (follow-up vs. baseline/pre-enrolment periods) was not considered. For example, a participant who underwent pre-enrolment surgery could be misclassified as having a new surgical event. Third, AI hallucinations can occur. Fourth, speech recognition errors from unrecognized pronunciations or transcription mistakes, for example, a Chinese word sounding like "dead", triggered false mortality prediction despite contextual evidence to the contrary.

### Comparison of Fu-LLM with GPT-4 with different prompts and different timepoints

To evaluate Fu-LLM (finetune_qwen2_7b), we performed a series of comprehensive baseline comparisons. First, Fu-LLM was compared with the benchmark public LLM GPT-4 with different prompts. We designed three prompts for GPT-4 to output the preadjudications, which were zero_shot, zero_shot_cot and one_shot_prompt (examples

**Table 1 | Baseline characteristics and endpoints of the study participants**

| Variables | Overall (n = 1022) | JL center (n = 307) | NFH center (n = 345) | JNU center (n = 370) |
|---|---|---|---|---|
| Demographics | | | | |
| Female, no (%) | 403 (39.4) | 106 (34.5) | 135 (39.1) | 162 (43.8) |
| Male, no (%) | 619 (60.6) | 201 (65.5) | 210 (60.9) | 208 (56.2) |
| Age, mean ± SD (years) | 63.1 (11.0) | 64.0 (11.3) | 62.7 (10.6) | 62.7 (11.1) |
| BMI, mean ± SD (kg/m$^2$) | 24.8 (3.6) | 24.4 (3.2) | 24.9 (3.3) | 25.1 (4.0) |
| Cardiac risk factors, no (%) | | | | |
| Hypertension | 599 (58.6) | 201 (65.5) | 209 (60.6) | 189 (51.1) |
| Diabetes | 235 (23.0) | 76 (24.8) | 94 (27.2) | 65 (17.6) |
| Dyslipidemia | 222 (21.7) | 75 (24.4) | 77 (22.3) | 70 (18.9) |
| Smoking | 167 (16.3) | 59 (19.2) | 38 (11.0) | 70 (18.9) |
| Family history of CAD | 65 (6.4) | 51 (16.6) | 5 (1.4) | 9 (2.4) |
| Symptoms, no (%) | | | | |
| Chest pain | 253 (24.8) | 85 (27.7) | 106 (30.7) | 62 (16.8) |
| Chest stuffiness | 365 (35.7) | 107 (34.9) | 141 (40.9) | 117 (31.6) |
| Dyspnea | 103 (10.1) | 36 (11.7) | 31 (9.0) | 36 (9.7) |
| Others (palpitation, syncope, etc.) | 204 (20.0) | 55 (17.9) | 80 (23.2) | 69 (18.6) |
| Asymptomatic | 350 (34.2) | 105 (34.2) | 89 (25.8) | 156 (42.2) |
| Degree of stenosis on CCTA, no (%) | | | | |
| CAD-RADS 2 | 437 (42.9) | 108 (35.4) | 144 (41.7) | 185 (50.1) |
| CAD-RADS 3 | 296 (29.0) | 99 (32.5) | 107 (31.0) | 90 (24.4) |
| CAD-RADS 4 | 286 (28.1) | 98 (32.1) | 94 (27.2) | 94 (25.5) |
| Group assignment, no (%) | | | | |
| CCTA + CT-FFR | 524 (51.3) | 155 (50.5) | 170 (49.3) | 199 (53.8) |
| CCTA alone | 498 (48.7) | 152 (49.5) | 175 (50.7) | 171 (46.2) |
| Recording files | | | | |
| Number of recording files, no (%) | 1046 (100) | 321 (30.7) | 355 (33.9) | 370 (35.4) |
| Duration of the recording (median (IQR), sec) | 147 (96–216) | 178 (127–258.5) | 177 (128–252) | 93 (63–136) |
| Length of the text (median (IQR), words) | 587.5 (362.8–865.5) | 713.0 (539.5–1034) | 750 (557–982) | 317.5 (238.0–499.5) |
| Outcomes[a], no (%) | | | | |
| 1-year ICA | 112 (11.0) | 45 (14.7) | 47 (13.7) | 20 (5.4) |
| 1-year revascularization | 92 (9.0) | 36 (11.7) | 40 (11.6) | 16 (4.3) |
| PCI | 82 (8.0) | 35 (11.4) | 35 (10.2) | 12 (3.2) |
| CABG | 10 (1.0) | 1 (0.3) | 5 (1.5) | 4 (1.1) |
| 1-year MACE | | | | |
| All cause death | 22 (2.2) | 11 (3.6) | 3 (0.9) | 8 (2.2) |
| Cardiac | 3 (0.3) | 1 (0.3) | 2 (0.6) | 0 |
| Noncardiac | 16 (1.6) | 9 (2.9) | 1 (0.3) | 6 (1.6) |
| Unknown | 3 (0.3) | 1 (0.3) | 0 | 2 (0.5) |
| Nonfatal acute MI | 5 (0.5) | 3 (1.0) | 1 (0.3) | 1 (0.3) |
| Urgent revascularization | 3 (0.3) | 0 | 3 (0.8) | 0 |

*BMI* body mass index, *CABG* coronary artery bypass grafting; *CAD* coronary artery disease; *CAD-RADS* coronary artery disease-reporting and data system; *CCTA* coronary CT angiography; *CT-FFR* CT derived fractional flow reserve; *JL* Jinling Hospital; *JNU* the affiliated hospital of Jining Medical University; *ICA* invasive coronary angiography; *MACE* major adverse cardiovascular event; *MI*, myocardial infarction; *NFH* Nanjing First Hospital; *PCI* percutaneous coronary intervention; *SD* standard deviation.
[a]The outcomes were adjudicated by an independently clinical events committee with referring to medical records (admission note or discharge summary at minimum), referred as golden reference standards.

are presented in Supplementary Method 3). Among these prompts, zero_shot achieved the highest overall agreement of 87.6% (95% CI [86.7–88.4], Cochran's $Q = 12.8$, $p = 0.002$) across all five outcomes, outperforming the one_shot prompt (86.8% [95% CI, 85.9–87.7], adjusted $p = 0.284$) and zero_shot_cot prompt (86.0% [95% CI, 85.1–87.1], adjusted $p = 0.001$). Zero_shot prompts yielded a sensitivity of 95.7% (95% CI, 94.9–96.6) and a specificity of 95.2% (95% CI, 94.4–96.0). The performances of the GPT-4 adjudications with the three kinds of prompts for each of the five outcomes are shown in Supplementary Table 3.

We subsequently conducted temporal drift testing for GPT-4 and compared Fu-LLM's performance with that of GPT-4 (zero shot) at three time points (September 2023; January 2024; April 2025) using data from the JL Center to assess whether the adjudications were consistent, evolved or deteriorated. The overall agreement, sensitivity and specificity significantly differed across the three time points

**Table 2 | Performances of Fu-LLM with different strategies for adjudications of clinical events**

| | Raw agreement, % (95% CI) | Sensitivity, % (95% CI) | Specificity, % (95% CI) | Positive predictive value, % (95% CI) | Negative predictive value, % (95% CI) |
|---|---|---|---|---|---|
| Performances of finetune_qwen2_7b | | | | | |
| Whether the information came from the participant himself/herself | 97.2 (96.1–98.2) | 97.9 (96.7–98.9) | 96.2 (94.5–97.9) | 97.4 (96.1–98.7) | 96.9 (95.2–98.6) |
| Whether the participant died | 99.8 (99.5–100.0) | 100.0 (100.0–100.0) | 99.8 (99.5–100.0) | 91.7 (79.3–100.0) | 100.0 (100.0–100.0) |
| Whether the participant was hospitalized[a] | 82.7 (80.4–85.0) | 88.9 (84.7–93.3) | 91.3 (88.9–93.6) | 79.1 (73.7–83.9) | 95.7 (94.0–97.3) |
| Whether the participant underwent surgery[a] | 92.1 (90.3–93.8) | 95.3 (91.8–100.0) | 89.8 (87.3–92.5) | 75.1 (69.5–80.5) | 98.4 (97.0–99.4) |
| Whether the participant taken medication[a] | 96.4 (95.2–97.5) | 99.8 (99.4–100.0) | 91.0 (86.1–95.5) | 98.5 (97.6–99.3) | 98.5 (96.2–100.0) |
| Total | 93.7 (93.1–94.3) | 97.5 (96.7–98.2) | 95.0 (94.2–95.8) | 93.1 (91.9–94.2) | 98.2 (97.8–98.7) |
| Performances of finetune_qwen2_7b_wo_aug | | | | | |
| Whether the information came from the participant himself/ herself | 92.7 (91.2–94.3) | 95.5 (93.8–97.1) | 88.7 (85.6–91.5) | 92.5 (90.5–94.6) | 93.1 (90.2–95.3) |
| Whether the participant died | 99.6 (99.2–99.9) | 100.0 (100.0–100.0) | 99.6 (99.2–99.9) | 84.6 (70.0–96.8) | 100.0 (100.0–100.0) |
| Whether the participant was hospitalized[a] | 69.5 (66.5–72.3) | 77.9 (72.6–83.1) | 89.8 (87.0–92.2) | 73.6 (67.9–79.8) | 91.7 (89.4–94.0) |
| Whether the participant underwent surgery[a] | 87.5 (85.5–89.5) | 87.7 (82.8–92.4) | 85.7 (82.8–88.7) | 66.4 (59.7–72.8) | 95.6 (93.7–97.4) |
| Whether the participant taken medication[a] | 94.8 (93.3–96.2) | 98.9 (98.2–99.5) | 81.4 (75.2–87.6) | 96.8 (95.4–98.0) | 92.9 (87.9–96.9) |
| Total | 88.9 (88.1–89.7) | 94.4 (93.3–95.4) | 92.1 (91.1–93.1) | 89.1 (87.7–90.5) | 96.0 (95.2–96.7) |
| Performances of zero_shot_qwen2_7b | | | | | |
| Whether the information came from the participant himself/ herself | 87.2 (85.0–89.0) | 91.2 (88.8–93.2) | 81.4 (77.7–85.0) | 87.8 (85.3–90.1) | 86.3 (82.6–89.3) |
| Whether the participant died | 99.4 (98.9–99.8) | 100.0 (100.0–100.0) | 99.4 (98.9–99.8) | 78.6 (63.0–91.7) | 100.0 (100.0–100.0) |
| Whether the participant was hospitalized[a] | 25.0 (22.4–27.5) | 51.4 (45.0–57.9) | 10.6 (8.0–13.1) | 17.5 (14.5–20.5) | 37.3 (29.9–44.9) |
| Whether the participant underwent surgery[a] | 67.3 (64.3–70.2) | 76.6 (69.9–82.6) | 64.4 (60.6–68.3) | 40.9 (35.7–46.3) | 89.5 (86.3–92.3) |
| Whether the participant taken medication[a] | 82.9 (80.5–85.2) | 83.6 (81.0–85.9) | 86.9 (81.3–92.1) | 97.3 (96.1–98.4) | 48.1 (42.3–54.2) |
| Total | 72.5 (71.3–73.7) | 82.1 (80.2–83.6) | 70.3 (68.5–71.9) | 65.5 (63.5–67.4) | 85.1 (83.5–86.5) |

*CI* confidence interval; *GPT* generative pretrained transformer.
[a]For participants who were reported as dead during follow-up, for humanitarian reasons, the follow-up staff would not inquire about the information of hospitalization, surgery or medication, therefore, these three events of the death cases would not be evaluated (22 recordings vignettes reported death events).

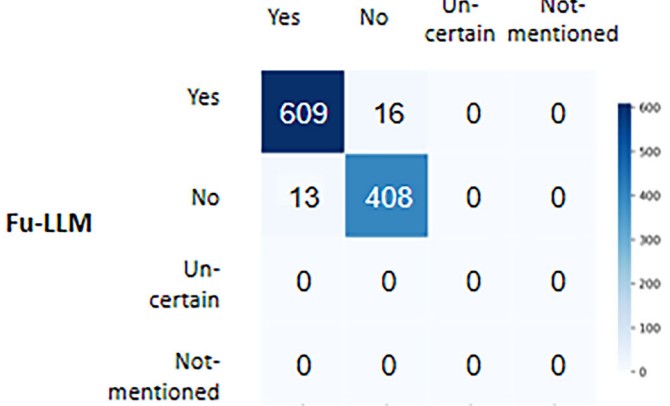

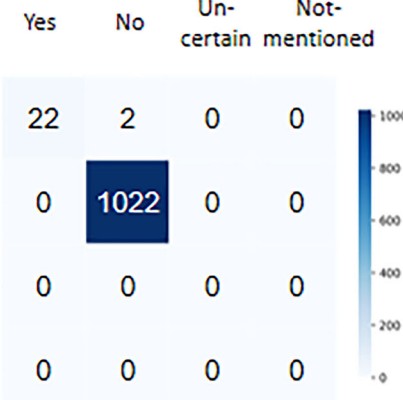

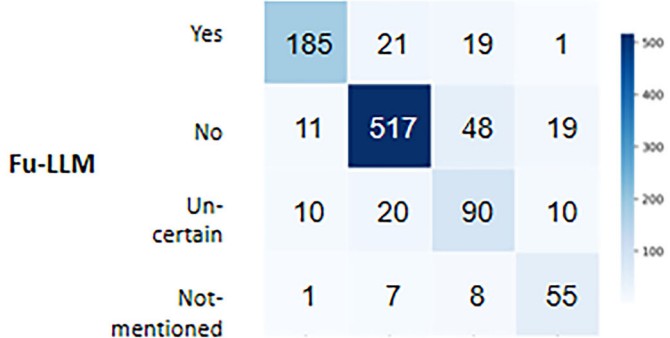

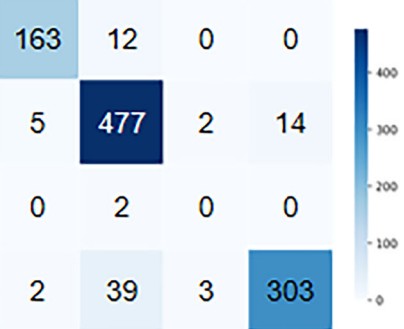

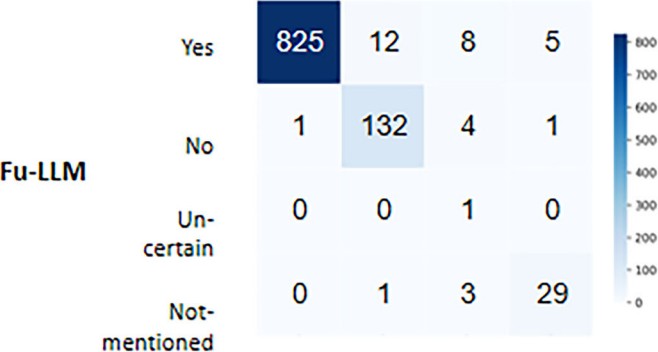

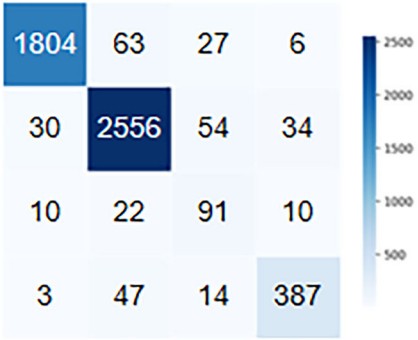

**Fig. 2 | Confusion matrices of the four adjudications for each outcome and all outcomes between Fu-LLM (finetune_qwen2_7b) and the silver references in the study dataset. A** Confusion matrices for the question of "whether the information came from the participant himself / herself". **B** Confusion matrices for the question of "whether the participant died during the follow-up period". **C** Confusion matrices for the question of "whether the participant was hospitalized during the follow-up period". **D** Confusion matrices for the question of "whether the participant underwent surgery during the follow-up period". **E** Confusion matrices for the question of "whether the participant took medication during the follow-up period". **F** Confusion matrices for all of the five questions. Source data are provided as a Source data file.

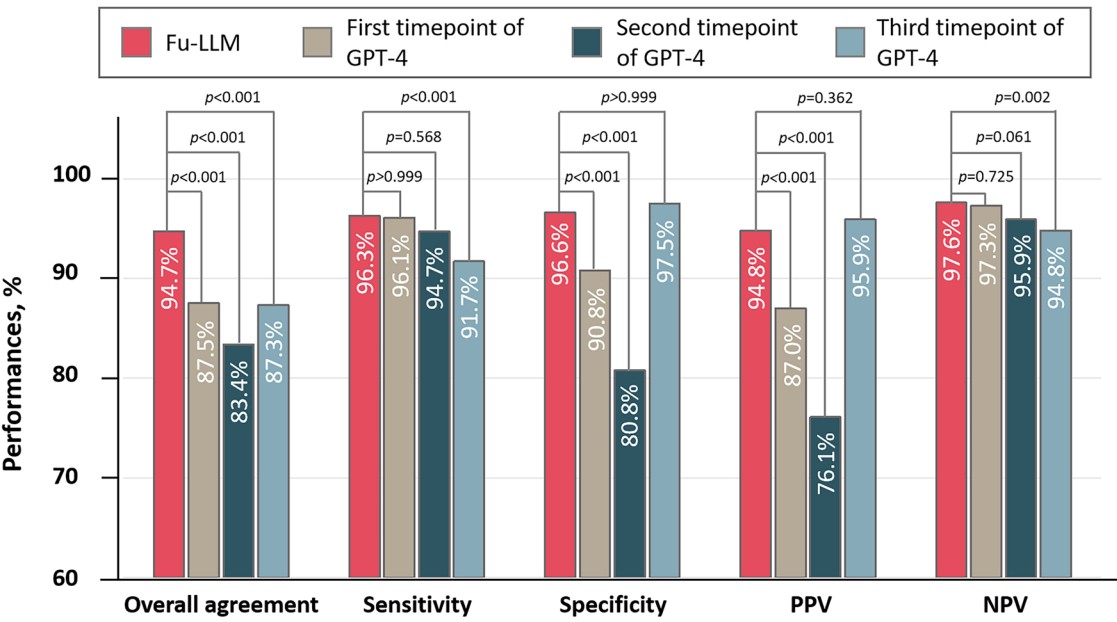

**Fig. 3 | Comparison of the performances between Fu-LLM (finetune_qwen2_7b) and GPT-4 at three different timepoints in the JL dataset.** Note: first timepoint of GPT-4 was September 2023; second timepoint of GPT-4 was January 2024; third timepoint of GPT-4 was April 2025. GPT generative pretrained transformer; NPV negative predictive value; PPV positive predictive value. Within-group differences of the overall agreement, sensitivity and specificity between Fu-LLM and GPT-4 at three different timepoints were assessed using Cochran's $Q$ statistic. For comparison of NPV and PPV to those of the GPT-4 at three different timepoints, $\chi^2$ test was applied. The statistical tests were two-sided with significance set at $p < 0.05$. $p$ had been adjusted by Bonferroni correction. Source data are provided as a Source data file.

(Cochran's $Q$ test, all $p < 0.001$; Supplementary Table 4). Fu-LLM demonstrated significantly higher overall agreement (94.7% [95% CI, 93.4–95.7]) than GPT-4 did at all time points (September 2023: 87.5% [95% CI, 85.8–89.1]; January 2024: 83.4% [95% CI, 81.6–85.2]; April 2025: 87.3% [95% CI, 85.6–88.9], all adjusted $p < 0.001$). The sensitivity of Fu-LLM was greater (96.3% [95% CI, 94.6–97.7]) than that at the April 2025 time point (91.7% [95% CI, 89.4–94.1], adjusted $p < 0.001$), with comparable performance to that at the September 2023 (96.1% [95% CI, 94.4–97.6], adjusted $p > 0.999$) and January 2024 (94.7% [95% CI, 92.7–96.7], adjusted $p = 0.824$) time points. The specificity of Fu-LLM (96.6% [95% CI, 95.3–97.7]) exceeded that at the September 2023 (90.8% [95% CI, 88.9–92.6], adjusted $p < 0.001$) and January 2024 (80.8% [95% CI, 78.1–83.4], adjusted $p < 0.001$) time points but matched that at the April 2025 time point (97.5% [95% CI, 96.4–98.4], adjusted $p > 0.999$) (Fig. 3 and Supplementary Table 4).

**Benchmark comparison with five other popular public LLM models.** Using the optimal prompt (zero_shot), we compared Fu-LLM with five other state-of-the-art public LLMs, namely, GPT-3.5-turbo (2025_01_25), GPT-4o (2024_11_20), DeepSeek-v3 (2024_12_26), claude 3.5-sonnet (2024_10_22) and gemini-2.0-pro (2025_02_05). These models were applied to clinical outcome adjudication, resulting in "yes," "no," "uncertain," or "not mentioned" classifications for all five outcomes. Fu-LLM demonstrated significantly greater overall agreement (Cochran's $Q = 731.4$; $p < 0.001$) and sensitivity (Cochran's $Q = 196.6$; $p < 0.001$) than all five LLMs did. Its specificity was comparable to those of GPT-3.5-turbo (93.3% [95% CI, 92.3–94.2], adjusted $p = 0.081$), GPT-4o (95.7% [95% CI, 95.0–96.5], adjusted $p > 0.999$), claude 3.5-sonnet (96.1% [95% CI, 95.3–96.8], adjusted $p > 0.999$), and gemini-2.0-pro (96.7% [95% CI, 96.0–97.4], adjusted $p = 0.115$) but exceeded that of DeepSeek-v3 (78.3% [95% CI, 76.8–79.8], adjusted $p < 0.001$). The performances of the five LLMs in terms of adjudications for each of the five outcomes are shown in Table 3; Supplementary Table 5 and Fig. 4.

### Development of a machine learning model for outcome preadjudication

To benchmark against traditional ML approaches, we implemented two conventional shallow models: SVM_TFIDF and SVM_Word2Vec. Both models followed the same data partitioning strategy followed in Fu-LLM using fivefold cross-validation. Word augmentation was applied to generate two additional models (SVM_TFIDF_wo-aug and SVM_Word2Vec_wo-aug). Details of the model development are provided in the Supplementary Materials. The SVM_Word2Vec model demonstrated overall agreement (81.3% [95% CI, 80.3–82.3] vs. 81.0% [95% CI, 79.9–82.0], adjusted $p > 0.999$), sensitivity (84.9% [95% CI, 83.2–86.6] vs. 83.5% [95% CI, 81.7–85.3], adjusted $p = 0.199$) and specificity (87.8% [95% CI, 86.4–89.0] vs. 86.8% [95% CI, 85.7–88.2], adjusted $p = 0.823$) that were comparable to those of SVM_TFIDF. Both models significantly outperformed SVM_TFIDF_wo-aug and SVM_Word2Vec_wo-aug (all adjusted $p < 0.001$). Compared with Fu-LLM, both SVM models showed inferior performance in terms of all the metrics (all adjusted $p < 0.001$). The performances of the four SVM models for each of the five outcomes are shown in Table 4, Supplementary Table 6 and Fig. 5.

### Comparison of the performances of Fu-LLM and human staff

Four follow-up staff members from the follow-up team of the Department of Radiology in Jinling Hospital were enrolled in the in silico human-model comparison study. They were radiology residents in training with 2 years of experience, had been trained for follow-up and had previously conducted approximately 30 telephone follow-up interviews. Staff independently adjudicated outcomes ("yes," "no," "uncertain," or "not mentioned") for all five outcomes using JNU Center data—drawn from an unfamiliar province—to avoid prior exposure. Compared with adjudication by Fu-LLM, adjudication by human staff showed significantly lower overall agreement (83.4% [95% CI, 82.6–84.3] vs. 92.3% [95% CI, 91.1–93.5]; $p < 0.001$), sensitivity (85.5% [95% CI, 84.1–86.8] vs. 97.5% [95% CI, 96.0–98.6], $p < 0.001$) and

**Table 3 | Total performances of five popular public LLMs for adjudications of clinical events**

| | Raw agreement, % (95% CI) | Sensitivity, % (95% CI) | Specificity, % (95% CI) | Positive predictive value, % (95% CI) | Negative predictive value, % (95% CI) |
|---|---|---|---|---|---|
| DeepSeek-v3 (2024_12_26) | 82.5 (81.5–83.5) | 94.4 (93.4–95.4) | 78.3 (76.8–79.8) | 75.0 (73.2–76.8) | 95.3 (94.4–96.2) |
| GPT-3.5-turbo (2025_01_25) | 82.5 (81.4–83.5) | 91.3 (90.1–92.6) | 93.3 (92.3–94.2) | 90.3 (89.0–91.6) | 94.0 (93.0–94.9) |
| GPT-4o (2024_11_20) | 85.7 (84.6–86.6) | 91.5 (90.1–92.7) | 95.7 (95.0–96.5) | 93.6 (92.5–94.7) | 94.2 (93.3–95.0) |
| claude 3.5-sonnet (2024_10_22) | 86.1 (85.1–87.0) | 95.7 (94.7–96.5) | 96.1 (95.3–96.8) | 94.4 (93.3–95.4) | 97.0 (96.3–97.6) |
| gemini-2.0-pro (2025_02_05) | 84.0 (83.0–85.0) | 93.2 (92.0–94.2) | 96.7 (96.0–97.4) | 95.1 (94.1–96.0) | 95.4 (94.6–96.2) |

*CI* confidence interval; *GPT* generative pretrained transformer.

specificity (87.0% [95% CI, 85.8–88.1] vs. 93.1% [95% CI, 91.3–94.7]; $p < 0.001$) across all events (Fig. 6 and Supplementary Table 7).

Considering that Fu-LLM had a high NPV (98.2% at JNU center), we evaluated the efficacy of a hypothetical hybrid human + Fu-LLM adjudication strategy, in which human investigators ruled out the negative adjudications identified by Fu-LLM for the 5 events and manually adjudicated the remaining cases. Therefore, for Fu-LLM adjudications of "yes" and "no", this strategy reduced the adjudication volume by half (57.2%, 848/1483), and follow-up staff only adjudicate outputs of "yes" (42.8%, 635/1483) while maintaining a sensitivity of 97.5 (573/588 true events captured), missing only 2.6% (16/588) of event-positive cases.

### Fu-LLM adjudications compared to those of the gold reference standards

Since outcomes of "death" and "ICA" were adjudicated by the CEC as the gold reference standard in the China CT-FFR study 3 using telephone interviews and medical records, we compared the silver references and Fu-LLM against the gold references. Both the silver references and Fu-LLM demonstrated substantial concordance for death events (raw agreement, 100.0% [95% CI, 100.0–100.0] and 99.8% [95% CI, 99.5–100.0]) with sensitivities of 100% and similar specificities (99.8% [95% CI, 99.4–100.0] vs 100% [95% CI, 100.0–100.0]). With respect to surgery events, the gold reference referred to ICA, and compared with Fu-LLM, the silver reference showed higher concordance (raw agreement, 56.4% [95% CI, 53.1–59.6] and 53.2% [95% CI, 50.0–56.2], $p < 0.001$) and specificity (55.3% [95% CI, 52.2–58.5] and 51.7% [95% CI, 48.2–55.1], $p < 0.001$), but comparable sensitivity (64.2% [95% CI, 56.0–72.5] and 65.0% [95% CI, 56.4–73.6], $p > 0.999$).

## Discussion

In this secondary analysis of a randomized clinical trial (China CT-FFR study 3), we developed a domain-specific large language model (Fu-LLM) for automated clinical outcome preadjudication from vignettes of follow-up telephone interviews using SFT with data augmentation (19,162 training samples). Fu-LLM achieved a raw agreement of 93.7%, a sensitivity of 97.5% and a specificity of 95.0% across all events. Comparative analyses demonstrated Fu-LLM's superiority over state-of-the-art LLMs (GPT-3.5-turbo, GPT-4o, DeepSeek-v3, Claude 3.5-Sonnet, Gemini-2.0-Pro) and traditional ML (SVM). Fu-LLM also outperformed human adjudicators in terms of agreement (92.3% vs. 83.4%; $p < 0.001$), sensitivity (97.5% vs. 85.5%; $p < 0.001$), and specificity (93.1% vs. 87.0%; $p < 0.001$). These results indicate the capacity of Fu-LLM to streamline clinical trial outcome curation through contextual interpretation of conversational data, representing a major advance in LLM applications for large-scale trial efficiency.

Conducting clinical trial follow-up via telephone interviews remains complex and resource-intensive, and requires cross-site collaboration, while follow-up staff at each site contact participants amidst demanding clinical workloads. LLMs demonstrate contextual tracking capabilities in conversations[13,14,25] and in-context learning for output alignment[26]. However, many language models have been utilized for "general purpose" without specializing in a specific domain type, and these models lack clinical specificity, leading to hallucinations in medical terminology and inconsistent performance across model versions. In contrast, domain-specific LLMs, such as Fu-LLM, overcome these limitations through task-targeted fine-tuning. In our study, we developed Fu-LLM—a domain-specific LLM for automated clinical outcome preadjudication—by specializing in follow-up dialog patterns and clinical event semantics using augmented clinical dialogs (>19k samples across three medical centers). Our findings underscore the value of task-specific adaptation and data augmentation for improving LLM performance in clinical outcome adjudication. Compared with the zero-shot baseline, fine-tuning the Qwen2-7B model on real-world follow-up dialog data significantly increased both the accuracy and consistency, highlighting the importance of aligning

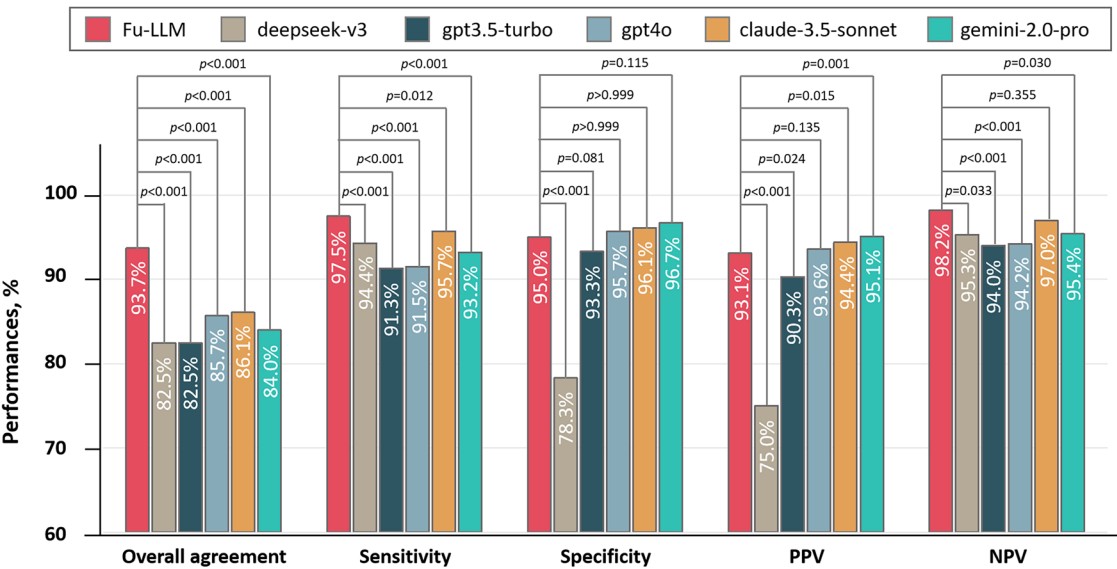

**Fig. 4 | Comparison of the performances between Fu-LLM (finetune_qwen2_7b) and five other popular public LLM models (DeepSeek-v3 (2024_12_26), GPT-3.5-turbo (2025_01_25), GPT-4o (2024_11_20), claude 3.5-sonnet (2024_10_22) and gemini-2.0-pro (2025_02_05)) in the study dataset.** GPT generative pretrained transformer; NPV negative predictive value; PPV positive predictive value. Within-group differences of the overall agreement, sensitivity and specificity between Fu-LLM and five other popular public LLM models were assessed using Cochran's Q statistic. For comparison of NPV and PPV to those of five other popular public LLM models, $\chi^2$ test was applied. The statistical tests were two-sided with significance set at $p < 0.05$. $p$ had been adjusted by Bonferroni correction. Source data are provided as a Source data file.

**Table 4 | Total performances of SVMs for adjudications of events with silver reference adjudications**

|  | Raw agreement, % (95% CI) | Sensitivity, % (95% CI) | Specificity, % (95% CI) | Positive predictive value, % (95% CI) | Negative predictive value, % (95% CI) |
|---|---|---|---|---|---|
| svm_w2v | 81.3 (80.3–82.3) | 84.9 (83.2–86.6) | 87.8 (86.4–89.0) | 82.7 (81.1–84.2) | 89.4 (88.2–90.6) |
| svm_w2v_wo_aug | 70.9 (69.6–72.2) | 78.3 (76.5–80.2) | 78.2 (76.6–79.6) | 71.2 (69.2–73.1) | 84.0 (82.5–85.4) |
| svm_tfidf | 81.0 (79.9–82.0) | 83.5 (81.7–85.3) | 86.8 (85.7–88.2) | 81.4 (79.5–83.0) | 88.4 (87.2–89.6) |
| svm_tfidf_wo_aug | 77.5 (76.4–78.7) | 81.5 (79.6–83.2) | 84.8 (83.5–86.1) | 78.7 (76.8–80.6) | 86.9 (85.7–88.2) |

*CI* confidence interval; *SVM* support vector machine.

general-purpose LLMs with domain-specific tasks. Furthermore, incorporating data augmentation techniques—particularly data rewriting and synthetic generation—yielded additional performance gains, likely by increasing data diversity and improving the model's generalizability to varied linguistic expressions. These results suggest that even modestly sized, carefully constructed training datasets, when combined with targeted augmentation, can substantially improve LLM utility in specialized clinical NLP tasks. Fu-LLM reached an overall high agreement of 93.7%, with a sensitivity and specificity greater than 95%, which are better than those of other general-purpose LLMs and traditional ML methods. Fu-LLM also outperformed a board of human staff.

This study has several implications for optimizing the workflow of clinical trials. First, automated adjudication of telephone interviews by Fu-LLM is a more resource-efficient alternative. With its high NPV, Fu-LLM reduced the workload by approximately 57.2% and could reduce the cost for large-scale clinical trials from telephone interviews while providing more precise outcome data for epidemiology research. Second, Fu-LLM enables quality control in multicentre follow-up studies by processing dialog within seconds, where the proportions of "uncertain" and "not mentioned" responses serve as interview quality indicators. Third, domain-specific LLMs such as Fu-LLM have potential for unaddressed processes—including trial design, site identification,

participant screening, and recruitment—although these applications require future investigation[27–29]. These findings position domain-specific LLMs as transformative tools for decentralized trials, where consistent, scalable endpoint adjudication is paramount. Revolutionizing clinical trials empowered by digital technology such as Fu-LLM is expected.

Fu-LLM may not always be correct, with four misclassification patterns and the potential for hallucinated responses being observed. In our study, the overall performance of Fu-LLM was greater than 90%, while the suboptimal performance on hospitalization adjudication (82.7%) stemmed primarily from participants' ambiguous descriptions of "health examinations" as hospital visits. For example, the participants may not indicate that they went to a hospital but may instead say "go to health examination", which made judgment confusing. In addition, the participants may say that they had a previous surgery, and the follow-up staff may subsequently determine that the participant was hospitalized for this surgery and would not further inquire whether he or she had been hospitalized, which the Fu-LLM could not do. Future prompts should explicitly distinguish routine check-ups from disease-related admissions. Nevertheless, Fu-LLM outperformed other public LLMs and demonstrated more consistent performance than GPT-4 did, whose outputs varied significantly across time points. In general, by balancing their reasoning strengths, Fu-LLM could be responsibly integrated into medical research and improve biomedical

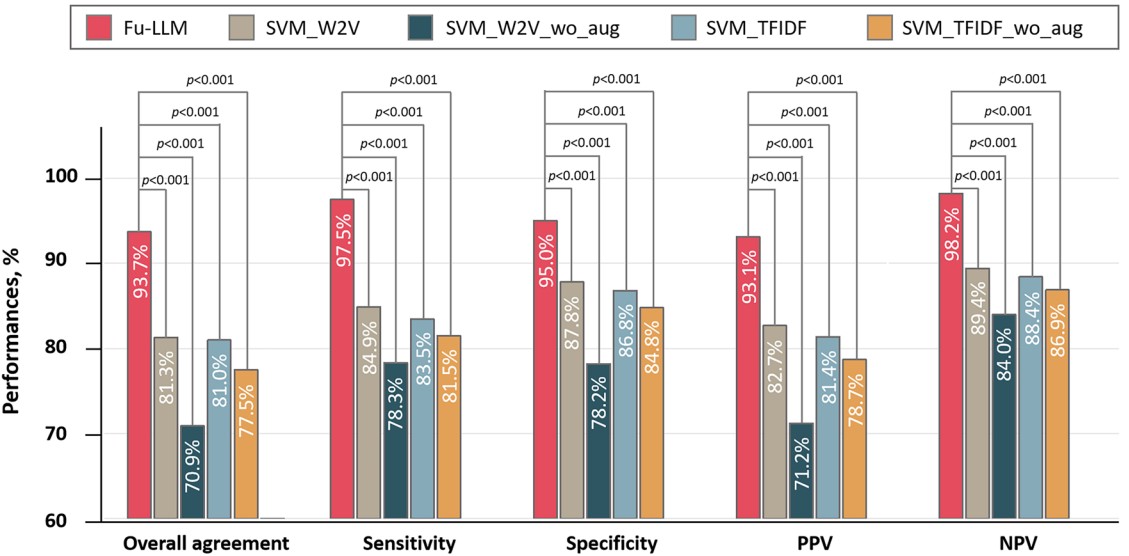

**Fig. 5 | Comparison of the performances between Fu-LLM (finetune_qwen2_7b) and the SVM models (SVM_TFIDF, SVM_TFIDF_wo_aug, SVM_Word2Vec and SVM_Word2Vec_wo_aug) in the study dataset.** NPV negative predictive value; PPV positive predictive value; SVM support vector machine; SVM_W2V, SVM_Word2-Vector. Within-group differences of the overall agreement, sensitivity and specificity between Fu-LLM and the SVM models were assessed using Cochran's $Q$ statistic. For comparison of NPV and PPV to those of the SVM models, the $\chi^2$ test was applied. The statistical tests were two-sided with significance set at $p < 0.05$. $p$ had been adjusted by Bonferroni correction. Source data are provided as a Source data file.

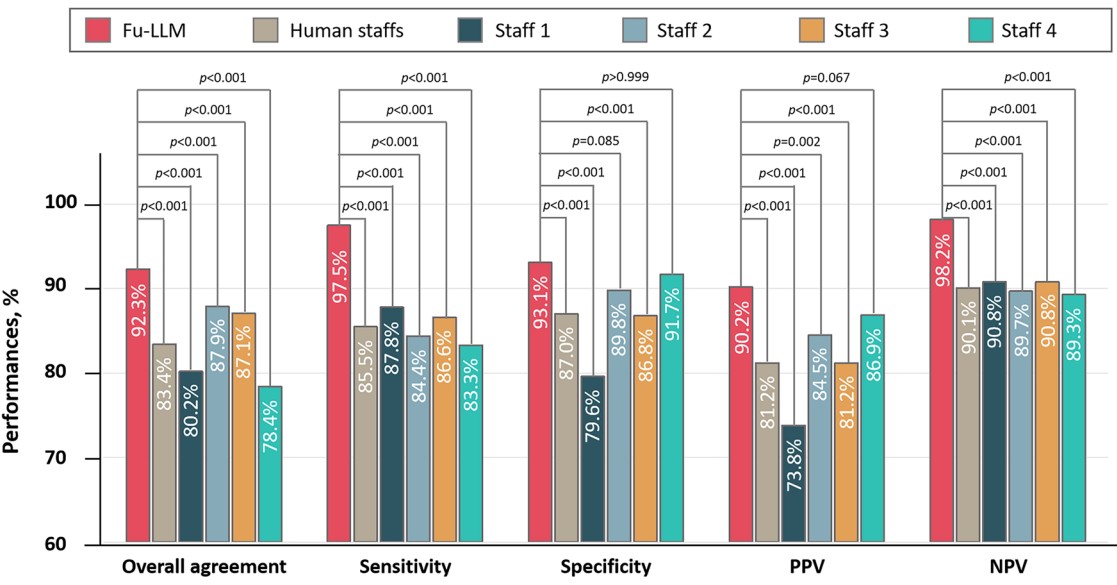

**Fig. 6 | Comparison of the performances between Fu-LLM (finetune_qwen2_7b) and the overall performance as well as each of the human staff in the JNU dataset.** Human staffs' adjudications demonstrated significantly lower overall agreement, sensitivity and specificity than those of Fu-LLM for all of the five events. NPV negative predictive value; PPV positive predictive value. Within-group differences of the overall agreement, sensitivity, specificity, NPV and PPV between Fu-LLM and the human staffs were assessed using $\chi^2$ test. Within-group differences of the overall agreement, sensitivity and specificity between Fu-LLM and each of the staffs were assessed using Cochran's $Q$ statistic. For comparison of NPV and PPV to those of each of the staffs, the $\chi^2$ test was applied. The statistical tests were two-sided with significance set at $p < 0.05$. $p$ had been adjusted by Bonferroni correction. Source data are provided as a Source data file.

research productivity and could ultimately promote health care equity.

This study has several limitations. First, our study was a secondary analysis of a randomized clinical trial, and biases could not be eliminated from the participants' selection. Second, the performance of follow-up staff augmented with Fu-LLM was not evaluated in this study, and combining Fu-LLM and human adjudication may be a practical approach. Third, Fu-LLM's inability to incorporate original audio

recordings constrained its adjudication ability. Given the progressive refinement in deep learning architectures, such as inception[30], this limitation may be addressed with future AI models. Fourth, the human adjudicators in our comparative analysis had limited follow-up experience (approximately 30 telephone interviews prior to this study). While this may impact direct performance comparisons with specialized teams, it reflects a common reality in large-scale multicentre trials where shortages of highly experienced follow-up

personnel are prevalent. The scarcity of domain experts necessitates the frequent involvement of less experienced staff in real-world clinical research settings. Fifth, despite Fu-LLM's superior efficiency, its missed event rate of 2.6% requires human oversight for critical outcomes such as mortality. Future integration with EHR autoalert systems could mitigate this limitation.

In conclusion, our study preliminarily verifies the feasibility of a domain-specific large language model (Fu-LLM) for clinical outcome adjudication from telephone interviews in the multi-centre China CT-FFR study 3 trial. Fu-LLM demonstrated greater agreement, sensitivity, specificity, and negative predictive value than both public state-of-the-art LLMs and traditional ML methods did. Automation of this process could ease the follow-up work burden placed on investigators because of its high sensitivity, negative predictive value and time-saving capability, which may highlight a pathway for the application of LLMs in future clinical trials. Although Fu-LLM shows significant promise, its deployment in clinical practice requires overcoming multiple barriers.

## Methods

### Ethics
The study was approved by the ethics committee of Jinling Hospital and has been registered (ChiCTR.org.cn Identifier: ChiCTR2400080585). This study followed all appropriate institutional and international guidelines and regulations for medical research, in line with the Declaration of Helsinki. This study constitutes a secondary analysis of the prospective China CT-FFR study 3 trial (ChiCTR.org.cn Identifier: ChiCTR2100053219) and utilizes exclusively anonymized data derived from the original dataset. No additional participant compensation was provided for this specific analysis, as no new patient contact or data collection occurred.

### Participants and recordings
This study is a secondary analysis of the China CT-FFR study 3 trial, and full details of the study protocol, including eligibility criteria, randomization, and intervention procedures, as well as 1-year outcome, have been published in a peer-reviewed journal[23]. Demographic information (gender, age, etc.), clinical information (hypertension, diabetes, hyperlipidaemia, smoking history, etc.), and CCTA-related interpretations were recorded. Ethics committees at the participating centers approved the trial protocol. All participants provided written informed consent. Follow-up was performed 90 days and 1 year after enrollment at the local study sites by telephone interviews and electronic medical records between February 28, 2022, and November 30, 2022. Telephone interviews were recorded during the 1-year follow-up. The primary endpoint of the study was the rate of ICA within 90 days after CCTA. Definitions of the primary and secondary outcomes can be found in Supplementary Method 2. This secondary analysis aimed to develop a domain-specific follow-up LLM (Fu-LLM) and investigate the feasibility of the model to automatically adjudicate participants' clinical outcomes from telephone interviews. Reporting followed the strengthening the reporting of observational studies in epidemiology reporting guidelines.

The inclusion criterion of the secondary study was participants who had completed a 1-year telephone interview follow-up with corresponding sound recordings saved in the clinical trial follow-up system ($n = 1191$). The exclusion criteria were as follows: (1) recordings with a participant who was unwilling to respond ($n = 27$); (2) failed recordings ($n = 34$); (3) poor recording quality, such as strong background noise affecting judgment ($n = 62$); and (4) recordings that were unrecognizable due to heavy local dialect and unclear pronunciation ($n = 22$).

### Construction of the silver reference standards
A consensus for establishing the silver reference standard adjudications according only to telephone interview recordings was developed by experts from the core laboratory and CEC of China CT-FFR study 3, which included 4 choices for each outcome: "yes", "no", "uncertain", and "not mentioned" (Supplementary Method 1). "uncertain" means that there was no clear "yes" or "no" judgment, and "not mentioned" means that the conversation did not contain information about the outcomes. Identifying information, such as the participant's name was removed.

The silver reference standards for this study were established by a panel of three follow-up professionals (J.Z., Z.C., and C.Y., who had conducted telephone follow-up interviews for more than 2 years and conducted more than 300 telephone interviews in 3 clinical trials). They were trained to adjudicate the outcomes according to the consensus and were required to independently listen to the recordings and determine the silver references. Each case was read by two annotators, and for cases that could not be definitively judged and those with discordant adjudications, they were marked and handed over to an expert evaluation panel for consultation and decision-making. Three annotators achieved substantial agreement (Fleiss' $\kappa = 0.86$) prior to expert arbitration, with discordant cases (8.1%, 420/5164) resolved by consensus discussion.

### Follow-up large language model (Fu-LLM) training
We conducted SFT based on the Qwen2-7B model to develop a domain-specific Fu-LLM that can automatically extract five specified clinical outcomes from follow-up dialogs. A total of 1046 real-world samples underwent fivefold cross-validation. Given the limited data size, we employed two data augmentation strategies to increase the data volume and diversity and implemented postprocessing techniques to further improve the quality of the generated data. Below, we describe the data augmentation and post-processing methods, followed by the training details.

Data augmentation strategies. 1) Data rewrite: we used the Qwen2-7B model to identify key dialog segments that directly or indirectly determine clinical outcomes. These critical segments were retained while approximately half of the nonessential dialog content was randomly removed, creating new training samples. This key-content clipping strategy helps the model focus on outcome-relevant dialog, thereby enhancing training efficiency. Prompt used for this method is provided in Supplementary Method 5. 2) Data synthesis: we leveraged existing follow-up dialogs as examples to generate new dialogs and corresponding clinical outcomes using Qwen2-7B. This approach expands the dataset and enhances diversity. The newly generated data underwent quality control and validation to ensure clinical plausibility and relevance. Prompt is provided in Supplementary Method 6.

Data postprocessing. We implemented an automatic two-step postprocessing pipeline using Qwen2-7B to reduce redundancy and improve data quality. 1) Data deduplication: to reduce data redundancy and mitigate the risk of overfitting during model training, we performed a strict exact-match deduplication on all generated data. This procedure ensures the uniqueness of each entry in the dataset by identifying and removing samples with identical textual content. 2) Content filtering: to address the tendency of LLM to append non-core content (such as conversational asides) to their outputs, we employed a set of heuristic rules for filtering. This process removes samples containing specific procedural descriptions (e.g., "has been deleted") or formatting anomalies (e.g., consecutive line breaks). By doing so, it purifies the dataset to consist solely of core dialog, thereby reinforcing the model's ability to learn key features.

Following 20 rounds of data augmentation, all samples were subjected to the post-processing pipeline described above. Ultimately, this strategy yielded 19,162 valid data entries, with 2439 originating

from the data rewrite strategy and 16,723 from the data synthesis strategy.

Fine-tuning Details. We employed SFT combined with low-rank adaptation (LoRA)[24] to fine-tune the Qwen2-7B model. The model was trained with a maximum sequence length of 4096 tokens and a gradient accumulation step of 8. Training was conducted on 6 NVIDIA H20 GPUs with a total batch size of 48. We used the AdamW optimizer with an initial learning rate of $1 \times 10^{-4}$, a cosine learning rate scheduler, and a warm-up ratio of 5% to stabilize the early training phase. The model was trained for 3 epochs to balance stability and performance.

### Machine learning models training

Two classical shallow models: SVM + TF-IDF and SVM + Word2Vec were developed. Both models followed the same data partitioning strategy as Fu-LLM, using five-fold cross-validation. Text preprocessing was standardized using the Jieba segmentation tool (https://github.com/fxsjy/jieba). In the SVM + TF-IDF method, we extracted up to 5000 features using a TF-IDF vectorizer and fed them into a multi-output SVM classifier with an RBF kernel to achieve multi-task prediction. In the SVM + Word2Vec method, a 100-dimensional Word2Vec embedding model was trained on the full dataset. Each document was represented as the average of its word embeddings and then input into the same multi-output SVM classifier.

### Comparison of Fu-LLM and human staff for outcomes adjudication

Four follow-up staff members from the follow-up team of the Department of Radiology in Jinling Hospital (H.P., H.W., Y.W. and J.Z., all of whom were radiology residents in training with 2 years of experience, and previously completed approximately 30 telephone follow-up interviews) were included. JNU Center recordings were selected for comparison because they originated from another province and were unfamiliar to the staffs. They were instructed to adjudicate the five outcomes independently into one of the four choices: "yes", "no", "uncertain", and "not mentioned" in the core laboratory according to the prespecified consensus.

### Comprehensive baseline comparisons between Fu-LLM and others

To evaluate the effectiveness of the proposed Fu-LLM, we conducted a series of comprehensive baseline comparisons:

1. Prompt optimization: three prompts were designed for GPT-4 to determine the optimal version, with the best-performing prompt compared with Fu-LLM.
2. Temporal drift testing: three GPT-4 versions (2023.09, 2024.01, 2025.04) were evaluated to assess performance variations across iterations.
3. State-of-the-art LLM comparison: Fu-LLM was benchmarked against five leading LLMs, including GPT-3.5-turbo (2025_01_25), GPT-4o (2024_11_20), DeepSeek-v3 (2024_12_26), claude 3.5-sonnet (2024_10_22) and gemini-2.0-pro (2025_02_05).
4. Traditional ML benchmark: two conventional models (SVM + TF-IDF and SVM + Word2Vec) were implemented using Fu-LLM's fivefold cross-validation strategy.
5. Human-model comparison: four follow-up staff members participated in the in-silico human-model comparison study.

### Statistical analysis

Numbers of "yes", "no", "uncertain", and "not mentioned" adjudications were tabulated for each outcome for all included cases for LLMs and human staffs according to the silver reference standards. For adjudications of "yes" and "no" by the silver reference adjudications, sensitivity, specificity, PPV, NPV, and 95% CI (using 1000-sample bootstrapping) for these statistics were reported. The performances

were also assessed in subgroups of each center and different group assignment (CCTA + CT-FFR group vs CCTA group). Inter-rater agreement was quantified using Fleiss' kappa for all outcomes, for all annotators who established the silver references. Multivariable logistic regression analysis was performed, and odds ratios were calculated. To measure the level of heterogeneity between Fu-LLM to other LLMs, Cochran's $Q$ statistic or McNemar analysis was used. For comparison of NPV, PPV, differences in subgroups, and the average performances of the four follow-up staffs to those of Fu-LLM, $\chi^2$ test was applied, against to silver references. For multiple group comparisons, statistical significance was evaluated with Bonferroni correction to control the family-wise error rate. For that the outcomes of death and ICA were finally determined by CEC in the CHINA CT-FFR 3 study (gold reference standards) according to telephone interviews and medical records, we further compared the silver references and Fu-LLM to that of the gold references by $\chi^2$ test. $p < 0.05$ was considered statistically significant. Statistical analyses were performed using SPSS Statistics (version 22.0.0; IBM) and Python (version 3.7.6).

### Reporting summary

Further information on research design is available in the Nature Portfolio Reporting Summary linked to this article.

### Data availability

The raw data has been deposited in Figshare (https://doi.org/10.6084/m9.figshare.25498102). Source data are provided with this paper.

### Code availability

The code of Fu-LLM is deposited in Zenodo (https://doi.org/10.5281/zenodo.17221355) and is also available at GitHub (https://github.com/OmniMedAI/FuLLM).

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

## Acknowledgements

This study was supported by the National Science and Technology Major Project for the Prevention and Treatment of Cancer, Cardiovascular and Cerebrovascular Diseases, Respiratory and Metabolic Diseases (2024ZD0521700 for L.J.Z.), National Natural Science Foundation of China (82441019 for L.J.Z.), the Key Projects of the National Natural Science Foundation of China (82230068 for L.J.Z.) and the National Natural Science Foundation of China (82102155 for Z.S.). The funders played no role in study design, data collection, analysis and interpretation of data, or the writing of this manuscript.

## Author contributions

L.J.Z. initiated the project and the collaboration. L.J.Z., Z.S., and B.H. contributed to the study concept and design. Z.S., B.W., B.H., J.Z., Z.C., C.Y., B.G., Q.X., and J.X. contributed to data acquisition, analysis and interpretation of data. Z.L., F.Z., and Y.Y. contributed to data augmentation, models development and validation. H.P., H.W., Y.W. and J.Z. contributed to the human-AI comparison study. L.J.Z., Z.S., and B.W. contributed to the drafting of the manuscript. Z.S. contributed to statistical analysis. L.J.Z. and Z.S. contributed to obtained funding. L.J.Z contributed to administrative, technical, material support and supervision.

## Competing interests

The authors declare no competing interests.
