## [Transparent Peer Review File · Nature Communications]

A Large Language Model for Clinical Outcome Adjudication from Telephone Follow-up Interviews: A Secondary Analysis of a Multicenter Randomized Clinical Trial

Corresponding Author: Professor Long Jiang Zhang

Version 0:

Reviewer comments:

Reviewer #1

(Remarks to the Author)

KEY RESULTS

- GPT-4 has superior performance to humans when adjudicating telephone follow-up for a RCT.

Validity

Overall the study is well conducted and validated across multiple sites and multiple time frames against a silver standard pre-screening and human gold standard re-review. Well done.

Originality and Significance

The fundamental premise of the study is that GPT4 can be utilized to automate adjudication of clinical trial follow-up calls. There are other early studies of this topic (see preprint reference below), this is the first that I know to be submitted for peer review. The use case is compelling, and the results encouraging.

Where other prompts attempted? A sensitivity analysis based on various prompting strategies could further expand the results.

Are there any patterns for the samples that GPT4 gets wrong? A detailed error analysis may prove insightful.

Data and Methodology

Fig.4 is a little hard to understand. I'd recommend labelling each sub-figure with the question that it is addressing so readers aren't forced to read through the legend to understand what they're looking at.

Fig. 5 is missing a legend.

The methods are straightforward and utilize prompting of GPT-4 to obtain structured outputs. The paper could benefit from additional technical work, particularly benchmarking against other LLM solutions in a similar zero-shot manner as well as finetuned against the paper's tasks.

The regressions to look for differential performance in the concordance are interesting, but not unexpected.

Appropriate use of statistics

Appropriate use of statistics where relevant.

Conclusions

The conclusions are somewhat expected and limited in the manuscript's current form. It is not surprising that GPT4, when appropriately prompted, is able to perform this task in a zero-shot manner. The results would be significantly strengthened with additional benchmarks or comparisons.

Suggested improvements

See above

References

Cunningham JW, Singh P, Reeder C, Claggett B, Marti-Castellote PM, Lau ES, Khurshid S, Batra P, Lubitz SA, Maddah M, Philippakis A, Desai AS, Ellinor PT, Vardeny O, Solomon SD, Ho JE (2023) Natural Language Processing for Adjudication of Heart Failure Hospitalizations in a Multi-Center Clinical Trial. medRxiv : the preprint server for health sciences, <https://doi.org/10.1101/2023.08.17.23294234>

Clarity and context

The writing could be substantially improved. Almost every paragraph has numerous areas with poor English grammar, "Telephone interviews follow-up is the most common applied method for its capacity on the rapid uptake of information and communication, which has demonstrated to yield comparable or lower levels of missing data, improved follow-up rate and clinical outcomes" for example.

Fig. 2's purpose is unclear and the chart is hard to interpret. Consider revising the legend or the chart.

Assessment of my expertise and areas that might be out of scope

None, I am an expert on this topic.

Reviewer #2

(Remarks to the Author)

In this study, the authors applied ChatGPT-4 to classify follow-up telephone interviews into 3 predefined categories including: 'yes', 'no', 'uncertain', and 'not mentioned', for 5 outcomes including (1) if the information was from the participant, (2) whether the participant died during the follow-up period, (3) whether the participant had hospitalized, (4) whether the participant had ICA, (5) whether the participant used medication. The authors first evaluated ChatGPT using a silver reference standard, manually determined by experts from the core laboratory and CEC of CHINA, only using telephone interviews. Next, the authors evaluated ChatGPT-4 using a gold standard determined by experts using both the follow-up interviews and the electronic health records. The evaluation shows that ChatGPT-4 has a good raw agreement of 92.5% with the silver standard.

The contribution of this study is very limited and some important information developing the silver and gold standards is missing. Specifically:

1. It's not clear how the silver reference and gold reference standards were determined. The authors simply reported that they were derived using domain experts, but it's not clear how many experts performed the determination, what's the criteria to determine each category, and what is the determination agreement among the domain experts.
2. I'm not surprised by the reported performance as determining the 5 outcome categories are typical text classification tasks that usually has very good performance, even without using large language models. Previous studies using traditional machine learning models such as support vector machines have reported very high performance for text classification. Determining the 5 outcomes from follow-up interviews is also not hard for humans.
3. The contribution of this study is very limited. It looks like the authors created the prompts and threw the documents to ChatGPT and ChatGPT solved everything. What's your contribution to this study? There are already tons of papers like "ChatGPT for XXX".
4. It's not clear how the authors conducted the experiment, whether the follow-up interviews contain privacy information, if there is a HIPAA-compliant environment such as Azure was used, as none of them were reported.

Reviewer #3

(Remarks to the Author)

I admit that I wasn't sure how to interpret Figure 2, the alluvial diagram. Some additional explanation might be helpful.

Figure 3 implies that the ChatGPT adjudications were used as an input for the silver references, but that's not what I understood from the text of the paper. Please clarify that.

You show how the adjudication results compared between ChatGPT and the silver and gold standards, but not how the various adjudication methods impacted the study's results. It might be interesting to report the differences between treatment arms using the different methods, limiting the analysis to just those patients where all three methods are available.

The fact that the results deteriorated when re-adjudicated after 3 months is worrisome. Do you have any sense for how this lack of reproducibility will be seen by regulatory agencies if this approach is used in a trial that's meant for regulatory submission?

Version 1:

Reviewer comments:

Reviewer #1

(Remarks to the Author)

All of my inquiries have been adequately answered. Love the current state of the work.

(Remarks on code availability)

The codebase is pretty thin and could use improving. Ideally it would include detailed instructions + files in order to replicate the work as-is. Currently the codebase is more of a rough sketch of what was done.

Reviewer #2

(Remarks to the Author)

This study was improved after revision. Specifically, the study team provided detailed information for the silver standard data construction. The study was redesigned to explore state-of-the-art LLMs, including Qwen and an advanced tuning algorithm based on LoRA; a comparison of LLMs with a traditional machine learning-based classifier, SVMs. The new experiments demonstrated improved performance of LLMs compared with traditional machine learning models. As the new model is based on an open-source LLM, Qwen, which is accessible to other researchers, it overcomes the limitation of ChatGPT, which is a closed-release model that researchers do not have direct access to reuse the proposed methods. Additional information about the protection of patient privacy and IRG approval was provided. Overall, the study was greatly improved. I have no further concern.

(Remarks on code availability)

The code on GitHub is not enough for others to replicate the study. Some of the new experiments are not reflected in the code.

Reviewer #3

(Remarks to the Author)

Thanks for your response to my comments.

(Remarks on code availability)

June 29, 2025

Regarding: Revision of manuscript

Dear referees,

We would like to express my thanks to the reviewers for their helpful and insightful comments for our manuscript (**NCOMMS-24-18840**) entitled “**Large Language Model Can Adjudicate Clinical Outcomes from Telephone Follow-up interviews: A Secondary Analysis of a Multicenter Randomized Clinical Trial**”, which have significantly improved the manuscript. We have taken all comments seriously and carefully revised the manuscript according to the suggestions.

To comprehensively address your suggestions—particularly regarding model robustness, comparative benchmarking, and error analysis—we significantly expanded the scope of our work. This required substantial time for ethical approvals, data reprocessing, model development, and rigorous validation, including:

1. **Developing a domain-specific LLM (Fu-LLM)** with supervised fine-tuning and data augmentation strategies (19,146 training samples);
2. **Benchmarking against 5 state-of-the-art LLMs** (GPT-4o, Claude 3.5, etc.) and traditional ML models (SVM variants);
3. **Conducting temporal stability tests** across 3 GPT-4 versions (2023 – 2025 releases);
4. **Implementing detailed error pattern analysis** (hallucinations, temporal confusion; Supplementary Material).

Our detailed responses to the specific comments are presented in the next pages. The original comments are in a red italic font, and our responses are in a black regular font. We are looking forward to getting further advice and comments, for us to achieve an acceptable version of the manuscript.

Again, thank you very much and look forward to hearing from you soon!

Sincerely yours,

Long Jiang Zhang, M.D., Ph.D.

Department of Radiology, Jinling Hospital, Affiliated Hospital of Medical School, Nanjing University. Nanjing, 210002, China

Reviewer #1 (R1) (Comments to the Author):

R1-1

1. KEY RESULTS

- GPT-4 has superior performance to humans when adjudicating telephone follow-up for a RCT.

Validity

Overall the study is well conducted and validated across multiple sites and multiple time frames against a silver standard pre-screening and human gold standard re-review. Well done.

Reply:

Thank you for your comments.

R1-2

2. Originality and Significance

The fundamental premise of the study is that GPT4 can be utilized to automate adjudication of clinical trial follow-up calls. There are other early studies of this topic (see preprint reference below), this is the first that I know to be submitted for peer review. The use case is compelling, and the results encouraging.

Reply:

Thank you for your comments. We also noticed the studies you mentioned, in which they developed a transformer-based natural language processing (NLP) model to adjudicate heart failure hospitalization [1,2]. While they can only adjudicates heart failure hospitalization from structured medical examinations (e.g. discharge summary text), and not natural dialogues. Large language model (LLM) was reported to have lower accuracy to interpret conversational formats compared to structured medical examinations [3].

Therefore, to address this fundamental concern, we have implemented the following key innovations in the revised manuscript to improve our study:

Firstly, our study developed a domain-specific large language model (Fu-LLM), which is the first LLM fine-tuned specifically for unstructured clinical dialogues, overcoming limitations of prior NLP models designed for structured texts. Fu-LLM was trained on 19,146 augmented dialogues (Section: "Development of a Domain-optimized LLM-based Model") using Qwen2-7B + LoRA. Fu-LLM designed for automatic clinical outcomes preadjudication from vignettes of follow-up telephone interviews, and they represented different kinds of researches. We have discussed the differences in Introduction Section in the revised manuscript.

Besides, we conducted a series of comprehensive baseline comparisons, which demonstrated Fu-LLM's superiority over state-of-the-art LLMs (GPT-3.5-turbo, GPT-4o, DeepSeek-v3, Claude 3.5-Sonnet, Gemini-2.0-Pro) and traditional machine learning (SVM). We conducted a temporal drift testing for GPT-4, in which the performances of Fu-LLM were compared with those of GPT-4 with zero-shot at three different timepoints (September 2023; January 2024; April 2025). Fu-LLM had significantly higher overall agreement than all of the GPT-4 with different timepoints. Fu-LLM also outperformed human adjudicators in agreement (92.3% vs. 83.4%; $p < 0.001$), sensitivity (97.5% vs. 85.5%; $p < 0.001$), and specificity (93.1% vs.

87.0%; $p < 0.001$). Fu-LLM enables >50% workload reduction in multicenter trials while maintaining a sensitivity of 97.5% (Section: "Comparison of the Performances Between Fu-LLM and Human Staffs"). The study established a new paradigm for automating clinical trial endpoints, far beyond applying generic LLMs, and provided a blueprint for translating foundation models into regulated clinical workflows.

References

1. Cunningham, J. W., et al. Natural language processing for adjudication of heart failure in the electronic health record. *JACC Heart Fail.* **11**, 852-854. <https://doi.org/10.1016/j.jchf.2023.02.012> (2023).
2. Cunningham, J. W., et al. Natural language processing for adjudication of heart failure in a multicenter clinical trial: A secondary analysis of a randomized clinical trial. *JAMA Cardiol.* **9**, 174-181. <https://doi.org/10.1001/jamacardio.2023.4859> (2024).
3. Johri, S., et al. An evaluation framework for clinical use of large language models in patient interaction tasks. *Nat Med.* **31**, 77-86. <https://doi.org/10.1038/s41591-024-03328-5> (2025).

R1-3

3. Where other prompts attempted? A sensitivity analysis based on various prompting strategies could further expand the results.

Reply:

Thank you for your insightful comments. We have added a sensitivity analysis, in which comparison of different prompts test for the proposed Fu-LLM was conducted in the revised draft, which included `finetune_qwen2_7b`, `finetune_qwen2_7b_wo_aug` and `zero_shot_qwen2_7b`. And the results demonstrated that `finetune_qwen2_7b` had the highest performances (93.7% [95% confidence intervals (CI), 93.1-94.3]; Cochran's $Q=1378.8$, $p < 0.001$, Table 2 in the revised manuscript). The three prompts are as follows:

- `finetune_qwen2_7b`, fine-tuned using all training dataset, including augmented samples;
- `finetune_qwen2_7b_wo_aug`, fine-tuned using original data without data augmentation;
- `zero_shot_qwen2_7b`, directly utilized the pre-trained Qwen2-7B model without any task-specific fine-tuning.

Besides, we also designed three prompts for GPT-4 to output the preadjudications and compared their differences, including zero-shot prompt, `zero_shot_cot` prompt and `one_shot` prompt (examples were shown in **Supplementary method 3** in the revised manuscript). Among the three prompts, `zero_shot` had the highest overall agreement of 87.6% (95%CI [86.7-88.4], Cochran's $Q=12.8$, $p=0.002$) for all of the five outcomes (Section: "Comparison of Fu-LLM with GPT-4 with Different Prompts and Different Timepoints"). Fu-LLM demonstrated significantly higher overall agreement than GPT-4 at all timepoints (all adjusted $p < 0.001$).

R1-4

4. Are there any patterns for the samples that GPT4 gets wrong? A detailed error analysis may prove insightful.

Reply:

Thank you for your thoughtful comments. We have added the description of the four patterns for the samples that Fu-LLM, the new proposed domain-specific LLM, gets wrong, which included the following four aspects: Firstly, overly long dialogues hindered information processing, while very brief dialogues provided insufficient detail. Secondly, failure to distinguish event timing (follow-up vs. baseline/pre-enrollment periods). For example, a participant with pre-enrollment surgery could be misclassified as having a new surgical event. Thirdly, AI hallucinations could happen. Fourthly, speech recognition errors from unrecognized pronunciations or transcription mistakes, for example, a Chinese word sounding like "dead" triggered false mortality prediction despite contextual evidence to the contrary. We also provided the corresponding examples for the wrong patterns in the **Supplementary Materials** in the revised manuscript (Section: "Risk Factors of Fu-LLM for Outcomes Preadjudication"), which would provides investigators with actionable intelligence to optimize Fu-LLM deployment — particularly for high-stakes outcomes like mortality and hospitalization adjudication.

R1-5

5. Data and Methodology

Fig.4 is a little hard to understand. I'd recommend labelling each sub-figure with the question that it is addressing so readers aren't forced to read through the legend to understand what they're looking at.

Reply:

Thank you for your comments. We have modified Fig.4 (in the original version) and labeled each sub-figure with the question in the revised manuscript (now Fig.2 in the revised version).

R1-6

Fig. 5 is missing a legend.

Reply:

Thank you for your comments. We have modified Fig.5 (in the original version) and added a legend for it (now Fig.6 in the revised version).

R1-7

7. The methods are straightforward and utilize prompting of GPT-4 to obtain structured outputs. The paper could benefit from additional technical work, particularly benchmarking against other LLM solutions in a similar zero-shot manner as well as fine-tuned against the paper's tasks

Reply:

Thank you for your insightful comments. In the revised manuscript, we have compared the performances of Fu-LLM to the benchmark public LLM of GPT-4 with different prompts. We designed three prompts for GPT-4 to output the preadjudications, which were zero_shot, zero_shot_cot and one_shot_prompt (examples were shown in **Supplementary method 3** in the revised manuscript). Among the three prompts, zero_shot had the highest overall agreement (87.6%, 95%CI [86.7-88.4], Cochran's Q=12.8, $p=0.002$) among the three prompts. And we also conducted a temporal drift testing for GPT-4, in which the performances of Fu-LLM were compared with those of GPT-4 with zero-shot at three different timepoints

(September 2023; January 2024; April 2025) for data from JL center to see whether the adjudications were consistent, evolved or deteriorated (**Fig. R1**). Fu-LLM had significantly higher overall agreement than all of the GPT-4 with different timepoints (Section: "Comparison of Fu-LLM with GPT-4 with Different Prompts and Different Timepoints").

Fig. R1 | Comparison of the performances between Fu-LLM (finetune_qwen2_7b) and GPT-4 at three different timepoints in the study dataset.

Note: First timepoint of GPT-4 was September 2023; Second timepoint of GPT-4 was January 2024; Third timepoint of GPT-4 was April 2025.

GPT, Generative pretrained transformer; NPV, Negative predictive value; PPV, Positive predictive value.

p had been adjusted by Bonferroni correction.

Besides, based on the most optimal prompt (zero_shot), other five state-of-the-art public LLMs, including GPT-3.5-turbo (2025_01_25), GPT-4o (2024_11_20), DeepSeek-v3 (2024_12_26), claude 3.5-sonnet (2024_10_22) and gemini-2.0-pro (2025_02_05), were also implied to the task of adjudicating clinical outcomes and compared to Fu-LLM. Fu-LLM had significantly higher overall agreement (Cochran's Q=731.4, $p<0.001$) and sensitivity (Cochran's Q=196.6, $p<0.001$) than all of the five LLM models (Section: "Benchmark Comparison With Five Other Popular Public LLM Models") (**Fig. R2**).

Fig. R2 | Comparison of the performances between Fu-LLM (finetune_qwen2_7b) and five other popular public LLM models (DeepSeek-v3 (2024_12_26), GPT-3.5-turbo (2025_01_25), GPT-4o (2024_11_20), claude 3.5-sonnet (2024_10_22) and gemini-2.0-pro (2025_02_05)) in the study dataset.

GPT, Generative pretrained transformer; NPV, Negative predictive value; PPV, Positive predictive value.

p had been adjusted by Bonferroni correction.

R1-8

8. The regressions to look for differential performance in the concordance are interesting, but not unexpected.

Reply:

Thank you for your comments. We appreciate your acknowledgment of the regression analysis. In the revised manuscript (Section: "Risk Factors of Fu-LLM for Outcomes Preadjudication"), we expanded this analysis to systematically identify factors influencing adjudication concordance. For example:

Hospitalization event adjudication: JNU center was an independent risk factor for discordance (aOR=2.7, $p<0.001$), likely due to regional dialect variations affecting voice-to-text conversion.

Medication event: Smoking history (aOR=0.3, $p=0.032$) and baseline CAD-RADS 2 status (aOR=4.5, $p=0.016$) significantly impacted concordance, reflecting how patient characteristics may introduce variability in self-reported dialogue.

These findings align with clinical intuition—demographic and center-level heterogeneity naturally introduce variations in conversational patterns. However, we emphasize that Fu-LLM maintained robust overall performance (93.7% agreement) despite these factors, demonstrating its generalizability.

R1-9

9. Appropriate use of statistics

Appropriate use of statistics where relevant.

Reply:

Thank you for your comments.

R1-10**10. Conclusions**

The conclusions are somewhat expected and limited in the manuscript's current form. It is not surprising that GPT4, when appropriately prompted, is able to perform this task in a zero-shot manner. The results would be significantly strengthened with additional benchmarks or comparisons.

Reply:

Thank you for your insightful critique. While zero-shot GPT-4 achieved moderate success (87.6% agreement), we fundamentally restructured the study to demonstrate Fu-LLM's superior technical innovation and clinical utility.

General-purpose LLM like GPT-4 was not enough to interpret conversational formats compared to structured medical documentation [1]. And general-purpose LLMs face other critical barriers, such as clinical domain errors, which include hallucinations [2] and omissions from medical document summarisation [3,4]; and temporal instability, which indicates that short- and long-time temporal variations of LLM's output exist [5]; and validation gaps, which means a lack of sufficient benchmarking against ground references. Rigorous studies developing domain-specific fine-tuning LLM with clinical dialogue augmentation and rigorous examining its clinical effectiveness in clinical outcomes adjudication in follow-up telephone interviews are warranted. Therefore, in this revised manuscript, we have reorganized the manuscript, and developed a domain-specific large language model (Fu-LLM) for automatic clinical outcomes preadjudication from vignettes of follow-up telephone interviews, with supervised fine-tuning and data augmentation (19,146 training samples).

We also added additional benchmarks of comparisons as suggested, and listed as follows:

1. Prompt Optimization: Three prompts were designed for GPT-4 to determine the optimal version, with the best-performing prompt compared against Fu-LLM.

2. Temporal Drift Testing: Three GPT-4 versions (2023.09, 2024.01, 2025.04) were evaluated to assess performance variations across iterations.

3. State-of-the-Art LLM Comparison: Fu-LLM was benchmarked against five leading LLMs, including GPT-3.5-turbo (2025_01_25), GPT-4o (2024_11_20), DeepSeek-v3 (2024_12_26), Claude 3.5-sonnet (2024_10_22) and Gemini-2.0-pro (2025_02_05).

4. Traditional Machine Learning Benchmark: Two classical models (SVM+TF-IDF and SVM+Word2Vec) were implemented using Fu-LLM's five-fold cross-validation strategy.

5. Human-Model Comparison: Four follow-up staff members from Jinling Hospital's Radiology Department participated in in-silico evaluations.

All of the results indicated that Fu-LLM can facilitate the curation of outcomes of clinical trials by the ability to contextually appropriate responses in a conversational setting.

We significantly expanded the scope of our work, which has fundamentally enhanced the completeness, innovation, and clinical impact of this work.

We have modified Conclusion as follows:

"In conclusion, our study preliminarily verifies the feasibility of a domain-specific large

language model (Fu-LLM) for clinical outcomes adjudication from telephone interview in the multicenter China CT-FFR Study 3 trial. Fu-LLM demonstrated higher agreement, sensitivity, specificity, and negative predictive value than both public state-of-the-art LLMs and traditional machine learning methods. Automating this process could ease the follow-up work burden upon investigators by high sensitivity, negative predictive value and time-saving capability, which may highlight a pathway for application of large language models in the future clinical trials. Although Fu-LLM shows significant promise, deployment into clinical practice will require multiple barriers to be overcome.”

References

1. Johri, S., et al. An evaluation framework for clinical use of large language models in patient interaction tasks. *Nat Med*, **31**, 77-86. <https://doi:10.1038/s41591-024-03328-5> (2025).
2. Xu, Z., Jain, S., & Kankanhalli, M. Hallucination is inevitable: an innate limitation of large language models. *Computer Science: Computation and Language*. <https://doi.org/10.48550/arXiv.2401.11817> (2025).
3. Asgari, E., et al. A framework to assess clinical safety and hallucination rates of LLMs for medical text summarisation. *NPJ Digit Med*. **8**, 274. <https://doi:10.1038/s41746-025-01670-7> (2025).
4. Zhang, T., et al. Benchmarking large language models for news summarization. *Trans. Assoc. Comput Linguist*. **12**, 39 – 57. https://doi:10.1162/tacl_a_00632 (2024).
5. Ćirković, A., Katz, T. Exploring the potential of ChatGPT-4 in predicting refractive surgery categorizations: Comparative study. *JMIR Form Res*. **7**, e51798. Published 2023 Dec 28. <https://doi:10.2196/51798> (2023).

R1-11

11. Clarity and context

The writing could be substantially improved. Almost every paragraph has numerous areas with poor English grammar, “Telephone interviews follow-up is the most common applied method for its capacity on the rapid uptake of information and communication, which has demonstrated to yield comparable or lower levels of missing data, improved follow-up rate and clinical outcomes” for example.

Reply:

Thank you for your comments. We have carefully revised the language of the manuscript. We hope you will find our revised manuscript satisfactory.

R1-12

12. Fig. 2’s purpose is unclear and the chart is hard to interpret. Consider revising the legend or the chart.

Reply:

Thank you for your thoughtful comments. We have carefully revised the manuscript. Fig. 2 (in the original version) is removed from the revised manuscript.

R1-13

13. Assessment of my expertise and areas that might be out of scope

14. None, I am an expert on this topic.

Reply:

We sincerely thank you for affirming your comprehensive expertise in this domain. Your authoritative perspective has been invaluable in strengthening the methodological rigor and clinical relevance of our work. We have meticulously addressed all your technical and conceptual suggestions in the revised manuscript, and welcome any additional insights your expertise may identify.

Reviewer #2 (R2) (Comments to the Author):

In this study, the authors applied ChatGPT-4 to classify follow-up telephone interviews into 3 predefined categories including: “yes”, “no”, “uncertain”, and “not mentioned”, for 5 outcomes including (1) if the information was from the participant, (2) whether the participant died during the follow-up period, (3) whether the participant had hospitalized, (4) whether the participant had ICA, (5) whether the participant used medication. The authors first evaluated ChatGPT using a silver reference standard, manually determined by experts from the core laboratory and CEC of CHINA, only using telephone interviews. Next, the authors evaluated ChatGPT-4 using a gold standard determined by experts using both the follow-up interviews and the electronic health records. The evaluation shows that ChatGPT-4 has a good raw agreement of 92.5% with the silver standard.

R2-1

The contribution of this study is very limited and some important information developing the silver and gold standards is missing. Specifically:

1. It's not clear how the silver reference and gold reference standards were determined. The authors simply reported that they were derived using domain experts, but it's not clear how many experts performed the determination, what's the criteria to determine each category, and what is the determination agreement among the domain experts.

Reply:

Thank you for your valuable comments. We have now provided comprehensive details in the Methods section (Section: "Silver Reference Standards Establishment"). Firstly, three well-trained follow-up professionals with >2 years of experience independently adjudicated outcomes based on predefined criteria. Each case was required to read by two annotators, and for cases that cannot be definitively judged and those with discordant adjudications, they would be marked and handed over to expert evaluation panel for consultation and decision. Explicit rules for each outcome category (e.g., hospitalization vs. health examination) are defined in **Supplementary Method 1**. Three annotators achieved substantial agreement (Fleiss' $\kappa = 0.86$) prior to expert arbitration, with discordant cases (8.1%, 420/5164) resolved by consensus discussion.

Gold references for death and ICA events were determined by the Clinical Events Committee using medical records. The independent clinical events committee was blinded to randomized group information, CCTA images, and the CT-FFR report but not to other information (eg, clinical history and results of electrocardiographic and laboratory examinations). The clinical events committee (including four clinicians with 12-30 years of

experience and one radiologist with 20 years of experience) adjudicated all primary and secondary end points based on standardized, prospectively determined definitions [1]. We have now comprehensively specified the methodology for determining gold reference standards in the supplementary materials of the revised manuscript (**Supplementary Method 2**).

In this revised manuscript, we have fundamentally restructured the study to demonstrate Fu-LLM's superior technical innovation and clinical utility. Firstly, we developed a domain-specific large language model (Fu-LLM) for automatic clinical outcomes preadjudication from vignettes of follow-up telephone interviews, with supervised fine-tuning and data augmentation (19,146 training samples).

To evaluate the effectiveness of the proposed Fu-LLM, we conducted a series of comprehensive baseline comparisons:

1). Prompt Optimization: Three prompts were designed for GPT-4 to determine the optimal version, with the best-performing prompt compared against Fu-LLM.

2). Temporal Drift Testing: Three GPT-4 versions (2023.09, 2024.01, 2025.04) were evaluated to assess performance variations across iterations.

3). State-of-the-Art LLM Comparison: Fu-LLM was benchmarked against five leading LLMs, including GPT-3.5-turbo (2025_01_25), GPT-4o (2024_11_20), DeepSeek-v3 (2024_12_26), claude 3.5-sonnet (2024_10_22) and gemini-2.0-pro (2025_02_05).

4). Traditional ML Benchmark: Two classical models (SVM+TF-IDF and SVM+Word2Vec) were implemented using Fu-LLM's five-fold cross-validation strategy.

5). Human-Model Comparison: Four follow-up staff members from Jinling Hospital's Radiology Department participated in in-silico evaluations.

All of the results indicated that Fu-LLM can facilitate the curation of outcomes of clinical trials by the ability to contextually appropriate responses in a conversational setting. This work established a new paradigm for automating clinical trial endpoints, far beyond applying generic LLMs, and provided a blueprint for translating foundation models into regulated clinical workflows.

We significantly expanded the scope of our work, which has fundamentally enhanced the completeness, innovation, and clinical impact of this work.

References

1. Guo, B., Xing, W., Hu, C., et al. Clinical effectiveness of automated coronary CT-derived fractional flow reserve: A Chinese randomized controlled trial. *Radiology*. **313**:e233354. <https://doi.org/10.1148/radiol.233354> (2024).

R2-2

2. I'm not surprised by the reported performance as determining the 5 outcome categories are typical text classification tasks that usually has very good performance, even without using large language models. Previous studies using traditional machine learning models such as support vector machines have reported very high performance for text classification. Determining the 5 outcomes from follow-up interviews is also not hard for humans.

Reply:

Thank you for your comments. Our work addresses following gaps. Firstly, unlike structured medical notes (e.g., discharge summaries), conversational dialogues contain ambiguous phrasing (e.g., "health examination" vs. hospitalization) and temporal confusion (e.g., pre-enrollment vs. follow-up events). And large language model (LLM) has been reported to have lower accuracy to interpret conversational formats compared to structured medical examinations [1]. In contrast to prior studies using structured text [2,3], our Fu-LLM is the first model specifically fine-tuned for clinical telephone dialogues (Qwen2-7B + LoRA). When benchmarked against SVM models (Section: "Development of a Machine Learning Model for Outcomes Preadjudication"), Fu-LLM achieved 94.7% agreement vs. SVM's 81.3% ($\Delta = 13.4\%$, $p < 0.001$; Fig. 5 in the revised manuscript), proving LLMs' superiority for contextual understanding (**Fig. R3**).

Fig. R3 | Comparison of the performances between Fu-LLM (finetune_qwen2_7b) and the SVM models (SVM_TFIDF, SVM_TFIDF_wo_aug, SVM_Word2Vec and SVM_Word2Vec_wo_aug) in the study dataset.

NPV, Negative predictive value; PPV, Positive predictive value; SVM, Support Vector Machine; SVM_W2V, SVM_Word2Vector.

p had been adjusted by Bonferroni correction.

Secondly, we also conducted a in-silico Human-Model comparison study. Four follow-up staffs were required to adjudicate the five outcomes in JNU center, because they were from another province and the four staffs had not been encountered before. Human staffs' adjudications demonstrated significantly lower agreement (83.4% [95%CI, 82.6-84.3] vs 92.3% [95%CI, 91.1-93.5]; $p < 0.001$), sensitivity (85.5% [95%CI, 84.1-86.8] vs 97.5% [95%CI, 96.0-98.6], $p < 0.001$) and specificity (87.0% [95%CI, 85.8-88.1] vs 93.1% [95%CI, 91.3-94.7]; $p < 0.001$) than those of Fu-LLM for all of the five events (**Fig. R4**).

Fig. R4 | Comparison of the performances between Fu-LLM (finetune_qwen2_7b) and the overall performance as well as each of the human staff. Human staffs' adjudications demonstrated significantly lower overall agreement, sensitivity and specificity than those of Fu-LLM for all of the five events.

NPV, Negative predictive value; PPV, Positive predictive value.

p had been adjusted by Bonferroni correction.

Thirdly, telephone follow-ups in multicenter trials generate exponential data burdens. Automated adjudication of telephone interview by Fu-LLM may offer a more resource-efficient alternative (57.2% adjudication volume reduction while maintaining 97.5% sensitivity according to our study). Besides, Fu-LLM may be used for quality control of follow-up, especially for multicenter studies. Fu-LLM can process a dialogue within seconds, and the proportions of responds of "Uncertain" and "Not mentioned" can be indications for telephone interview quality.

In general, this reframing directly links conversational data challenges to clinical trial pain points, positioning Fu-LLM as an operational necessity—not just a technical novelty.

References

1. Johri, S., et al. An evaluation framework for clinical use of large language models in patient interaction tasks. *Nat Med.* **31**, 77-86. <https://doi.org/10.1038/s41591-024-03328-5> (2025).
2. Cunningham, J. W., et al. Natural language processing for adjudication of heart failure in the electronic health record. *JACC Heart Fail.* **11**, 852-854. <https://doi.org/10.1016/j.jchf.2023.02.012> (2023).
3. Cunningham, J. W., et al. Natural language processing for adjudication of heart failure in a multicenter clinical trial: A secondary analysis of a randomized clinical trial. *JAMA Cardiol.* **9**, 174-181. <https://doi.org/10.1001/jamacardio.2023.4859> (2024).

3. The contribution of this study is very limited. It looks like the authors created the prompts and threw the documents to ChatGPT and ChatGPT solved everything. What's your contribution to this study? There are already tons of papers like "ChatGPT for XXX".

Reply:

We profoundly thank you for prompting us to clarify Fu-LLM's transformative innovations. Our revised work transcends conventional prompt engineering through three paradigm-shifting contributions:

Firstly, Domain-Specific LLM Development, Not Just Prompting. In the revised manuscript, Fine-tuned Qwen2-7B with clinical dialogue-specific adaptations via Low-Rank Adaptation (LoRA) was developed. We also generated 19,146 synthetic dialogues via Data rewrite and Data synthesis, while other public LLMs (e.g. GPT-4) lack domain-specific tuning for clinical dialogue ambiguity resolution.

Secondly, Rigorous Validation Against Clinical and Technical Benchmarks. We demonstrated Fu-LLM's superiority through unprecedented comparisons: vs 6 state-of-the-art LLMs (94.7% vs. 82.5%–86.1%, average $\Delta > 10.3\%$, $p < 0.001$); vs traditional ML (13.4% overall agreement gain over best SVM model); temporal stability (Unlike fluctuating GPT-4 versions (accuracy range: 83.4-87.5%), Fu-LLM maintained 94.7% agreement across intervals); Clinical Grounding (Validated against Clinical Events Committee-adjudicated gold standards (death/ICA events)).

Thirdly, Deployable Workflow Innovation for Multicenter Trials. We solved tangible clinical inefficiencies by proposing Hybrid Human-AI Framework, in which Fu-LLM auto-adjudicated 57.2% of cases, and human experts focused only on complex cases (e.g., ambiguous "hospital visits"). This strategy would reduce adjudications volume by half (57.2%) and follow-up staff only adjudicate those with answer of "Yes" (42.8%), while remaining a sensitivity of 97.5%.

We thank the reviewer for prompting us to clarify how Fu-LLM's specialized architecture, rigorous validation, and deployable efficiency gains establish a new paradigm for automating clinical trial endpoints—far beyond applying generic LLMs. This work provides a blueprint for translating foundation models into regulated clinical workflows.

References

1. Johri, S., et al. An evaluation framework for clinical use of large language models in patient interaction tasks. *Nat Med*, **31**, 77-86. <https://doi:10.1038/s41591-024-03328-5> (2025).
2. Xu, Z., Jain, S., & Kankanhalli, M. Hallucination is inevitable: an innate limitation of large language models. *Computer Science: Computation and Language*. <https://doi.org/10.48550/arXiv.2401.11817> (2025).
3. Asgari, E., et al. A framework to assess clinical safety and hallucination rates of LLMs for medical text summarisation. *NPJ Digit Med*. **8**, 274. <https://doi:10.1038/s41746-025-01670-7> (2025).
4. Zhang, T., et al. Benchmarking large language models for news summarization. *Trans. Assoc. Comput Linguist*. **12**, 39 – 57. https://doi:10.1162/tacl_a_00632 (2024).
5. Ćirković, A., Katz, T. Exploring the potential of ChatGPT-4 in predicting refractive surgery categorizations: Comparative study. *JMIR Form Res*. **7**, e51798. Published 2023 Dec 28.

<https://doi:10.2196/51798> (2023).

R2-4

4. It's not clear how the authors conducted the experiment, whether the follow-up interviews contain privacy information, if there is a HIPAA-compliant environment such as Azure was used, as none of them were reported.

Reply:

Thank you for your comments. In this revised manuscript, we clarified the experiment details. We confirmed strict adherence to privacy protocols. All conversation texts were de-identified before model training (Section: "Study Design and Participants"). Voice-to-text conversions used HIPAA-compliant servers (iFlytek API). And the study was approved by Jinling Hospital Ethics Committee and had been registered (ChiCTR.org.cn Identifier: ChiCTR2400080585).

Reviewer #3 (R3) (Comments to the Author):

R3-1

1. I admit that I wasn't sure how to interpret Figure 2, the alluvial diagram. Some additional explanation might be helpful.

Reply:

Thank you for your comments. We have removed the original Fig. 2 (Alluvial diagram) due to interpretability concerns. It is replaced with Fig. 2: Confusion matrices, which clearly visualizes per-outcome performance against silver standards.

R3-2

2. Figure 3 implies that the ChatGPT adjudications were used as an input for the silver references, but that's not what I understood from the text of the paper. Please clarify that.

Reply:

Thank you for your comments. We apologize for the ambiguity, and the ChatGPT adjudications were not used as an input for the silver references. The details of silver reference standard establishment were provided in the Methods section (Section: "Silver Reference Standards Establishment"). We have removed the original Fig. 3 from the revised manuscript.

R3-3

3. You show how the adjudication results compared between ChatGPT and the silver and gold standards, but not how the various adjudication methods impacted the study's results. It might be interesting to report the differences between treatment arms using the different methods, limiting the analysis to just those patients where all three methods are available.

Reply:

Thank you for your comments. As suggested, we had added subgroup analysis, and compared outcomes between CCTA vs. CCTA+CT-FFR groups using Fu-LLM adjudications. Results showed no significant differences in raw agreement (93.4% vs. 93.9%, $p=0.441$),

sensitivity (96.9% vs. 98.1%, $p=0.090$) or specificity (94.9% vs. 95.1%, $p=0.839$) (Supplementary Table 2 in the revised manuscript).

R3-4

4. The fact that the results deteriorated when re-adjudicated after 3 months is worrisome. Do you have any sense for how this lack of reproducibility will be seen by regulatory agencies if this approach is used in a trial that's meant for regulatory submission?

Reply:

Thank you for your comments. In this manuscript, our developed Fu-LLM showed stable performance (94.7% agreement) across intervals, while GPT-4 fluctuated slightly (83.4-87.5%) (Fig. 3 and Supplementary Table 3 in the revised manuscript). Therefore, a domain-specific LLM is needed and the model version should be freezed with predefined drift-detection protocols. Besides, human-AI hybrid workflows (Section: "Discussion") to mitigate reproducibility risks may also work.

Oct 09, 2025

Regarding: Revision of manuscript

Dear referees,

We would like to express my thanks to the reviewers for their helpful and insightful comments for our manuscript (**NCOMMS-24-18840A**) entitled "**A Large Language Model for Clinical Outcome Adjudication from Telephone Follow-up Interviews: A Secondary Analysis of a Multicenter Randomized Clinical Trial**", which have significantly improved the manuscript. We have taken all comments seriously and carefully revised the manuscript according to the suggestions.

Our detailed responses to the specific comments are presented in the next pages. The original comments are in a red italic font, and our responses are in a black regular font. We are looking forward to getting further advice and comments, for us to achieve an acceptable version of the manuscript.

Again, thank you very much and look forward to hearing from you soon!

Sincerely yours,

Long Jiang Zhang, M.D., Ph.D.

Department of Radiology, Jinling Hospital, Affiliated Hospital of Medical School, Nanjing University. Nanjing, 210002, China

Reviewer #1 (R1) (Comments to the Author):

R1-1

1. All of my inquiries have been adequately answered. Love the current state of the work.

Reply:

We sincerely thank you for your time and insightful comments, which have greatly improved our manuscript.

R1-2

2. The codebase is pretty thin and could use improving. Ideally it would include detailed instructions + files in order to replicate the work as-is. Currently the codebase is more of a rough sketch of what was done.

Reply:

We appreciate your feedback regarding the need for a more comprehensive codebase. You rightly emphasized the importance of detailed implementation guidelines to ensure reproducibility.

In direct response to your suggestion, we have substantially enhanced our GitHub repository to include:

- **Environment & Data Preparation:** Detailed step-by-step instructions and scripts for setting up the environment, preprocessing data, and running inference.
- **Core Implementation Code:** We provide well-commented scripts for all experiments, specifically featuring:
 - Code for our data augmentation strategies (rewrite and synthesis).
 - Scripts for fine-tuning the Qwen2-7B model using LoRA.
 - Complete evaluation code for all baseline comparisons (commercial LLMs, SVM models, and human evaluators).
- **Documentation & Reproducibility:** We provide detailed materials to guarantee that every result and figure in the paper can be faithfully reproduced:
 - A comprehensive README.md documentation explains the repository's structure and dependencies.
 - Input files and a minimal working example to lower the barrier for replication.

The improved codebase is now publicly available at: [GitHub Repository Link: <https://github.com/OmniMedAI/FuLLM>], and the code is also deposited in Zenodo (<https://doi.org/10.5281/zenodo.17221355>). We are confident that these enhancements will allow researchers to replicate our work in its entirety.

Reviewer #2 (R2) (Comments to the Author):

R2-1

This study was improved after revision. Specifically, the study team provided detailed information for the silver standard data construction. The study was redesigned to explore state-of-the-art LLMs, including Qwen and an advanced tuning algorithm based on LoRA; a comparison of LLMs with a traditional machine learning-based classifier, SVMs. The new

experiments demonstrated improved performance of LLMs compared with traditional machine learning models. As the new model is based on an open-source LLM, Qwen, which is accessible to other researchers, it overcomes the limitation of ChatGPT, which is a closed-release model that researchers do not have direct access to reuse the proposed methods. Additional information about the protection of patient privacy and IRG approval was provided. Overall, the study was greatly improved. I have no further concern.

Reply:

We thank you for your positive assessment of our revisions and your recognition of our enhancements in silver standard construction, LLM comparisons, and ethical documentation.

R2-2

2. The code on GitHub is not enough for others to replicate the study. Some of the new experiments are not reflected in the code.

Reply:

We thank you for highlighting the importance of code completeness. We have now fully updated the code repository to reflect all experiments introduced during the revision, including:

- **Environment & Data Preparation:** Detailed step-by-step instructions and scripts for setting up the environment, preprocessing data, and running inference.
- **Core Implementation Code:** We provide well-commented scripts for all experiments, specifically featuring:
 - Code for our data augmentation strategies (rewrite and synthesis).
 - Scripts for fine-tuning the Qwen2-7B model using LoRA.
 - Complete evaluation code for all baseline comparisons (commercial LLMs, SVM models, and human evaluators).
- **Documentation & Reproducibility:** We provide detailed materials to guarantee that every result and figure in the paper can be faithfully reproduced:
 - A comprehensive README.md documentation explains the repository's structure and dependencies.
 - Input files and a minimal working example to lower the barrier for replication.

The repository has been thoroughly revised to ensure transparency and reproducibility. You may access it here: [GitHub Repository Link: <https://github.com/OmniMedAI/FuLLM>], and the code is also deposited in Zenodo (<https://doi.org/10.5281/zenodo.17221355>).

Reviewer #3 (R3) (Comments to the Author):

R3-1

Thanks for your response to my comments..

Reply:

We thank you for acknowledging our responses and are pleased that our revisions addressed your previous concerns satisfactorily.